# Generalized Out-of-Distribution Detection and Beyond in Vision Language Model Era: A Survey

**Atsuyuki Miyai** *miyai@cvm.t.u-tokyo.ac.jp*
*The University of Tokyo*

**Jingkang Yang** *jingkang001@ntu.edu.sg*
*S-Lab, Nanyang Technological University*

**Jingyang Zhang** *jz288@duke.edu*
*Duke University*

**Yifei Ming** *yifei.ming@salesforce.com*
*Salesforce AI Research*

**Yueqian Lin** *yueqian.lin@duke.edu*
*Duke University*

**Qing Yu** *yu@hal.t.u-tokyo.ac.jp*
*The University of Tokyo*
*LY Corporation*

**Go Irie** *goirie@ieee.org*
*Tokyo University of Science*

**Shafiq Joty** *sjoty@salesforce.com*
*Salesforce AI Research*
*Nanyang Technological University*

**Yixuan Li** *sharonli@cs.wisc.edu*
*University of Wisconsin–Madison*

**Hai Helen Li** *hai.li@duke.edu*
*Duke University*

**Ziwei Liu** *ziwei.liu@ntu.edu.sg*
*S-Lab, Nanyang Technological University*

**Toshihiko Yamasaki** *yamasaki@cvm.t.u-tokyo.ac.jp*
*The University of Tokyo*

**Kiyoharu Aizawa** *aizawa@hal.t.u-tokyo.ac.jp*
*The University of Tokyo*
*Tokyo University of Science*

**Reviewed on OpenReview:** *https://openreview.net/forum?id=FO3IA4lUEY*

## Abstract

Detecting out-of-distribution (OOD) samples is crucial for ensuring the safety of machine learning systems and has shaped the field of OOD detection. Meanwhile, several other problems are closely related to OOD detection, including anomaly detection (AD), novelty detection (ND), open set recognition (OSR), and outlier detection (OD). To unify these

problems, a generalized OOD detection framework was proposed, taxonomically categorizing these five problems. However, Vision Language Models (VLMs) such as CLIP have significantly changed the paradigm and blurred the boundaries between these fields, again confusing researchers. In this survey, we first present a generalized OOD detection v2, encapsulating the evolution of these fields in the VLM era. Our framework reveals that, with some field inactivity and integration, the demanding challenges have become OOD detection and AD. Then, we highlight the significant shift in the definition, problem settings, and benchmarks; we thus feature a comprehensive review of the methodology for OOD detection and related tasks to clarify their relationship to OOD detection. Finally, we explore the advancements in the emerging Large Vision Language Model (LVLM) era, such as GPT-4V. We conclude with open challenges and future directions. The resource is available at https://github.com/AtsuMiyai/Awesome-OOD-VLM.

# 1 Introduction

A reliable visual recognition system should not only accurately predict known contexts, but also identify and reject unknown examples (Amodei et al., 2016; Mohseni et al., 2021; Hendrycks et al., 2021b; Hendrycks & Mazeika, 2022). In critical applications such as autonomous driving, the system must alert and cede control to the driver upon encountering unfamiliar scenes or objects not seen during training. However, most existing machine learning models are trained based on the closed-world assumption (Krizhevsky et al., 2012; He et al., 2015), where the test data is assumed to be drawn *i.i.d.* from the same distribution as the training data, known as in-distribution (ID). Therefore, the development of classifiers capable of detecting out-of-distribution (OOD) samples is a crucial challenge for real-world applications. This challenge is precisely the focus of research in the field of OOD detection.

While OOD detection primarily focuses on semantic distribution shift, several other tasks share similar goals and motivations, including outlier detection (OD) (Aggarwal & Yu, 2001; Hodge & Austin, 2004; Ben-Gal, 2005; Wang et al., 2019a), anomaly detection (AD) (Ruff et al., 2021; Pang et al., 2021; Bulusu et al., 2020; Chalapathy & Chawla, 2019), novelty detection (ND) (Pimentel et al., 2014; Miljković, 2010; Markou & Singh, 2003a;b), and open set recognition (OSR) (Boult et al., 2019; Geng et al., 2020; Mahdavi & Carvalho, 2021). Subtle differences in the specific definitions among these sub-topics have caused confusion in the field, leading to similar approaches being proposed across them.

To address this issue, the generalized OOD detection framework was introduced (Yang et al., 2024). The taxonomy of the generalized OOD detection framework is shown in Fig. 1. The generalized OOD detection framework introduces a taxonomy built upon four criteria: distribution shift type (covariate/semantic shift), the type of ID data (single/multi-class), necessity of ID classification, and learning setting (transductive/inductive). According to the above taxonomy, these five problems can be clearly categorized as shown in Fig. 1: AD is categorized into sensory AD, which deals with covariate shift, and semantic AD, which deals with semantic shift. ND falls under the same category as semantic AD. In multi-class settings requiring ID classification, both OSR and OOD detection are included. OD belongs to a transductive category (*i.e.*, it has access to all observations). This framework provides clear definitions and fosters a deeper understanding of each field.

In recent years, the emergence of Vision Language Models (VLMs), represented by CLIP (Radford et al., 2021), has rapidly accelerated research in the field of computer vision. This has changed the paradigm of the recognition field, allowing for zero-shot (Radford et al., 2021) or few-shot learning (Zhou et al., 2022c;b) in various domains. VLMs have significantly influenced the aforementioned five problems (OD, AD, ND, OSR, and OOD detection), and the application of VLMs, particularly CLIP, has become a highly notable research field (Ming et al., 2022a; Jeong et al., 2023; Miyai et al., 2023b; Zhou et al., 2024a). However, alongside this remarkable progress, the paradigm shift with the advent of the VLMs has blurred the boundaries between the five problems. Due to the difficulty of a clear understanding of the distinctions and interrelations between these tasks, each community within the fields is facing significant challenges in identifying the optimal direction to pursue in this VLM era.

| Observation Type | Covariate Shift Detection | | Semantic Shift Detection | | ID Classification |
|---|---|---|---|---|---|
| | Single-Class | Multi-Class | Single-Class | Multi-Class | |
| Inductive | **(a) Sensory AD** | | **(b) Semantic AD /ND** | | Not Necessary |
| | | | | **(c) OCR** | Necessary |
| | | | | **(d) OOD Detection** | ※ Density-based OOD detection methods do not require ID classification |
| Transductive | **(e) OD** | | | | Not Necessary |

Figure 1: Taxonomy of generalized OOD detection framework (Yang et al., 2024), illustrated by classification tasks. (a) Sensory anomaly detection is categorized as covariate shift detection. (b) Semantic anomaly detection and novelty detection fall under semantic shift detection. (c) Open set recognition and (d) out-of-distribution detection are classified as multi-class semantic shift detection tasks that require ID classification. (e) Outlier detection is characterized by having a transductive observation type.

In this survey, we introduce a novel unified framework termed *generalized OOD detection v2*, which extends the previous generalized OOD detection framework and summarizes the evolution of these five problems in the VLM era. To create it, we systematically review the use of VLMs across these five problem areas, tracing their development from the start to the present, and summarize the evolutionary trajectory of each problem. Importantly, our framework reveals that a paradigm shift has caused some fields to become inactive or integrate with others, and the demanding challenges in the VLM era become AD and OOD detection, which is a remarkable finding for each community. In addition to the inter-field evolution, we elaborate on the important shifts in the definition of OOD detection as well as the problem settings and benchmarks, with the contrast of those for related tasks. Then, we conduct a thorough review of the methodology for OOD detection and related tasks in the VLM era, intending to clarify their similarities and differences and inspire future research in OOD detection.

Finally, we introduce the early evolution of these problems in the emerging Large Vision Language Model (LVLM) era, such as GPT-4V (OpenAI et al., 2023) or LLaVA (Liu et al., 2023a; 2024b). We summarize the definition of each evolving problem, the findings so far, and future challenges.

To summarize, this survey presents four contributions to the research field:

1. **Proposing a Generalized OOD Detection v2**: We analyze the progression of five related topics (AD, ND, OSR, OOD detection, and OD) in the VLM era and propose *generalized OOD detection v2*. Our framework reveals that the paradigm shift has led to some field inactivity or integration, and the demanding challenges are AD and OOD detection. We hope that these observations highlight the demanding challenges in the VLM era and foster collaborative efforts among each community.

2. **An Extensive Survey for VLM-based OOD Detection**: We provide a comprehensive survey of VLM-based OOD detection and AD methods. In particular, we examine recent advances in zero-shot and few-shot settings by categorizing methods according to their training strategies and the use of additional prompts. Although various surveys on OD, AD, ND, OSR, and OOD detection have been conducted (Ruff et al., 2021; Pang et al., 2021; Bulusu et al., 2020; Chalapathy & Chawla, 2019; Geng et al., 2020; Yang et al., 2024; Cao et al., 2024b; Liu et al., 2024c; Xu & Ding, 2025), this work is the

Table 1: Number of VLM-based papers in the Top Venues from 2021 to April 2025

| Task | Top Venue |
|---|---|
| (a) Sensory AD | CVPR2023×1 (Jeong et al., 2023), ICLR2024×1 (Zhou et al., 2024a), CVPR2024×4 (Li et al., 2024c; Ho et al., 2024; Zhu & Pang, 2024; Huang et al., 2024), ACMMM×2 (Gu et al., 2024a; Zhu et al., 2024a), IJCAI2024×1 (Zuo et al., 2024), ECCV2024×2 (Cao et al., 2024c; Qu et al., 2024), NeurIPS2024×3 (Li et al., 2024f; Arodi et al., 2024), ICLR2025×2 (Jiang et al., 2025; Yun et al., 2025), CVPR2025×6 (Sun et al., 2025; Ma et al., 2025; Qu et al., 2025; Zhang et al., 2025b; Fan et al., 2025; Gu et al., 2025) |
| (b) Semantic AD/ND | TMLR2022×1 (Liznerski et al., 2022), CVPR2024×1 (Zhu & Pang, 2024), ICLR2025×1 (Yun et al., 2025) |
| (c) OSR | ECCV2024×1 (Miller et al., 2024) |
| (d) OOD Detection | NeurIPS2021×1 (Fort et al., 2021), AAAI2022×1 (Esmaeilpour et al., 2022), NeurIPS2022×1 (Ming et al., 2022a), ICCV2023×1 (Wang et al., 2023a), IJCV2023×1 (Ming & Li, 2024a), NeurIPS2023×4 (Miyai et al., 2023b; Tu et al., 2023; Park et al., 2023; Liu et al., 2023b), ICLR2024×2 (Nie et al., 2023; Jiang et al., 2024), CVPR2024×2 (Bai et al., 2024a; Li et al., 2024b), ICML2024×1 (Cao et al., 2024a), ECCV2024×4 (Zhang et al., 2024d; Liu & Christopher, 2024; Zhang et al., 2024f; Lafon et al., 2024), NeurIPS2024×5 (Li et al., 2024e; Yu et al., 2024; Chen et al., 2024a; Zhang et al., 2024a; Zhang & Zhang, 2024), TMLR2024×1 (Adaloglou et al., 2024), IJCV2025×1 (Miyai et al., 2025b), ICLR2025×1 (Zeng et al., 2025) |
| (e) OD | ICML2024×1 (Liang et al., 2024) |

first to comprehensively review VLM-based OOD detection methods. By examining the connections among related tasks, we aim to provide insights that enhance the understanding of OOD detection.

3. **An Introduction to the Evolution in the LVLM Era**: We further introduce the evolution of each problem in the LVLM era. Despite the infant stage of these fields, this survey offers an in-depth introduction to each problem, aiming to facilitate future advancements in this area.

4. **Open Challenges and Future Directions**: Finally, we discuss open challenges and future research directions. In particular, by conducting a comparative analysis between the fields of AD and OOD, we identify key areas that are especially important for advancing VLM-based OOD detection. We hope that this survey will serve as a valuable reference for future research on OOD detection and related tasks in the VLM era.

In the VLM era, research on open-vocabulary segmentation (Gu et al., 2022) and referring segmentation (Wang et al., 2022b) has been related to the discovery of unseen classes. However, these fields mainly focus on tasks that aim to generalize to examples from unseen classes during training when the class name is already known. In contrast, OOD detection and related fields deal with entirely new data, where even the class name is unknown. The goal of OOD detection and related fields is to detect them instead of generalizing to them. Therefore, due to this fundamental difference in motivation and objective, we put the tasks such as open-vocabulary segmentation outside the scope of this survey.

The paper content is organized as follows. In Sec. 2, we introduce the new version of generalized OOD detection by summarizing the evolution of the five related fields in the VLM era. We then overview the two key problems (OOD detection and AD) that have evolved and remain active in Sec. 3, with a detailed breakdown of existing methodologies being presented in Sec. 4 (VLM-based OOD detection) and Sec. 5

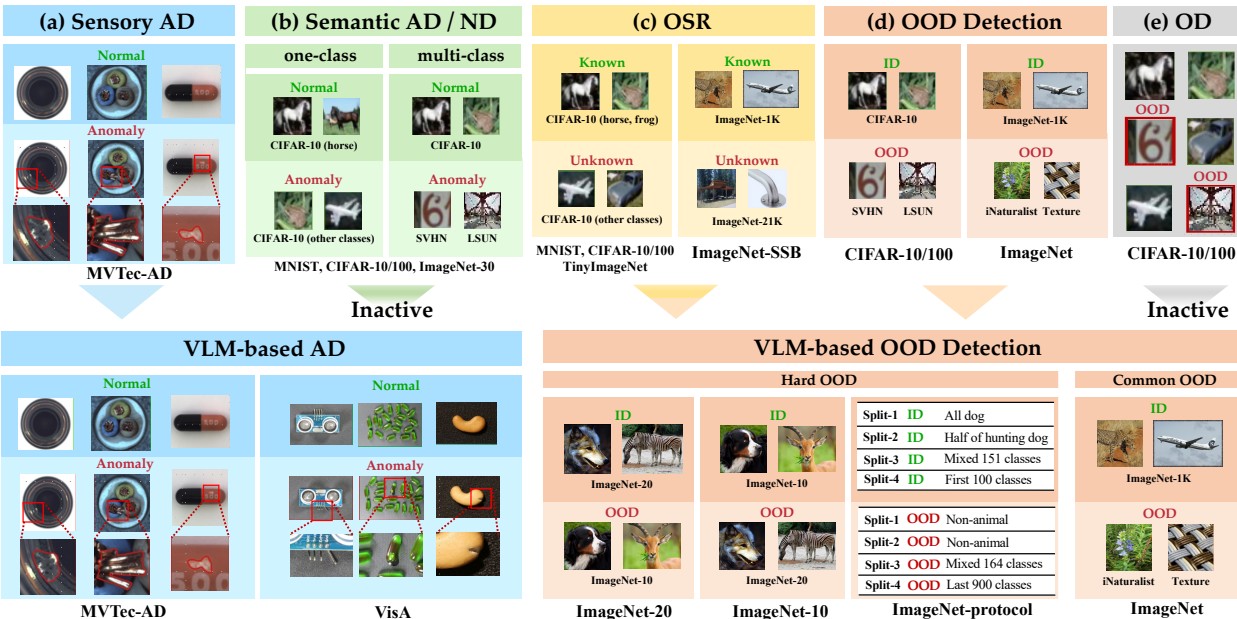

Figure 2: Generalized OOD detection framework v2, reflecting the evolution of each problem in the VLM era. (a) Sensory AD has consistently been an active research field even after the emergence of VLMs. In terms of benchmarks, in addition to the commonly used MVTec-AD (Bergmann et al., 2019), VisA (Zou et al., 2022), the largest industrial anomaly detection dataset, has also become a standard benchmark in the field. (b) Semantic AD/ND has become inactive in the VLM era. (c) OSR has been integrated into hard OOD detection. VLM-based hard OOD detection incorporates the benchmark setup of OSR and creates new benchmarks such as ImageNet-10/ImageNet-20 (Ming et al., 2022a) and ImageNet-protocol (Palechor et al., 2023; Li et al., 2024b). (d) OOD detection is a highly active research area in the VLM era. (e) OD has become inactive in the VLM era.

(VLM-based AD). In Sec. 7, we introduce early advancements of OOD detection and AD in the LVLM era. Sec. 8 features future directions. Finally, we conclude with Sec. 9.

## 2    Generalized OOD Detection V2

In this section, we introduce a novel unified framework termed *generalized OOD detection v2*, which summarizes the evolution of the five related fields in the VLM era. We first revisit the previous generalized OOD detection framework in Sec. 2.1. Next, in Sec. 2.2, we show the results of the investigation of the research activity in each field. In Sec. 2.3, we introduce the evolution of each problem. Finally, in Sec. 2.4, we have a discussion about the future directions.

### 2.1    Background: Generalized OOD Detection V1

We first briefly revisit a previous *generalized OOD detection*, which encapsulates five related sub-topics: anomaly detection (AD), novelty detection (ND), open set recognition (OSR), out-of-distribution (OOD) detection, and outlier detection (OD). These sub-topics can be similar in the sense that they all define a certain *in-distribution*, with the common goal of detecting *out-of-distribution* samples under the open-world assumption. Previously, subtle differences existed among the sub-topics in terms of the specific definition and properties of in-distribution (ID) and OOD data.

To provide a clear definition, a generalized OOD detection framework was proposed (Yang et al., 2024). The taxonomy for generalized OOD detection is shown in Fig. 1. It is based on the following four bases: (1) Distribution shift to detect: The task focuses on detecting either covariate shift (*e.g.*, OOD samples from a

different domain) or semantic shift (*e.g.*, OOD samples from a different semantic). (2) ID data type: The in-distribution (ID) data contains either a single class or multiple classes. (3) Whether the task requires ID classification: Some tasks require classification of the ID data, while others do not. (4) Transductive vs. inductive learning: Transductive tasks require all observations (both ID and OOD), while inductive tasks follow the common train-test scheme. According to the above taxonomy, these five problems can be clearly categorized as shown in Fig. 1: Anomaly detection is categorized into sensory anomaly detection, which deals with covariate shift, and semantic anomaly detection, which deals with semantic shift. Novelty detection falls under the same category as semantic anomaly detection. When addressing a multi-class scenario that necessitates ID classification, both open-set recognition and out-of-distribution detection are encompassed within this category. The main difference between OSR and OOD detection was the benchmark setup (Yang et al., 2024; Salehi et al., 2022) (Sec. 2.3 (c)). Outlier detection belongs to a different category from the other tasks, as this problem is transductive (*i.e.*, it has access to all observations).

The more precise definitions of these tasks are as follows:

**Anomaly Detection** Anomaly detection (AD) focuses on identifying anomalous instances during inference that deviate from a predefined notion of normality (Chandola et al., 2009). These anomalies can arise from two distinct types of distributional changes: covariate shift and semantic shift (Ruff et al., 2021). Accordingly, AD can be categorized into two primary sub-problems: sensory AD and semantic AD. Sensory AD targets anomalies caused by covariate shift, under the assumption that all normal instances are drawn from the same distribution. In this setting, no semantic shift is considered. In contrast, semantic AD addresses label shift, where normalities are assumed to come from a fixed semantic category. The objective in this setting is to detect test samples that belong to novel, previously unseen classes.

**Novelty Detection** Novelty detection (ND) aims to identify test samples that do not belong to any of the categories seen during training. Depending on the number of training classes, ND can be divided into two settings: One-class novelty detection, where the training set contains only one class, and multi-class novelty detection, where multiple classes are present in the training data. It is important to note that, even when multiple ID classes are available, the goal of multi-class novelty detection is solely to distinguish novel samples from known ones, without requiring classification among the ID classes. Both one-class and multi-class ND are binary classification tasks.

**Open Set Recognition** Open set recognition (OSR) requires a multi-class classifier to perform two tasks at the same time: (1) correctly classify test samples that belong to known classes, and (2) detect test samples that come from unknown classes. Here, it is common practice to refer to OOD classes as unknown classes and ID classes as known classes. However, there is no difference in meaning between these terms.

**Out-of-Distribution Detection** Out-of-distribution (OOD) detection aims to identify test samples that come from a distribution different from the training distribution. In most machine learning tasks, it typically refers to the label distribution, meaning that OOD samples should have labels not seen in the training data. It is also important to note that the training set usually includes multiple classes, and OOD detection should not degrade the model's ability to classify ID samples correctly.

**Outlier Detection** Outlier detection (OD) aims to identify samples that significantly differ from the rest of the given observation set. These differences may arise from either covariate shift or semantic shift. OD is often used as a pre-processing step for downstream tasks such as learning from open-set noisy labels (Wang et al., 2018) and open-set semi-supervised learning (Yu et al., 2020).

## 2.2 Investigation of Research Activity in Each Field

To investigate the research activity in each field, we comprehensively investigated papers that utilize VLMs from top venues and summarized them in Table 1. In this context, the term "top venues" refers to leading conferences characterized by high impact factors and rigorous peer-review standards, including NeurIPS, AAAI, ICLR, CVPR, ICML, ICCV, ECCV, IJCAI, and ACMMM, together with distinguished journals of comparable impact and recognition, such as TPAMI, IJCV, and TMLR. The authors visited the official pages of each venue and manually counted the number of VLM-based papers by examining their titles and content. Since CLIP, the most representative VLMs, was introduced at ICML 2021, we focused our investigation on

the period from 2021 to April 2025. We define the research activity of a field objectively by the number of papers published through a rigorous peer-review process in the period starting with the introduction of VLMs (*e.g.*, CLIP) and continuing to the present.

Our survey reveals that CLIP (Radford et al., 2021) is by far the most widely used VLM for OOD detection, whereas other models such as Grounding DINO (Liu et al., 2024e) and SAM (Kirillov et al., 2023) have seen limited adoption in this context. While many existing methods are developed with CLIP, the concept of OOD detection with CLIP is not limited to this particular model. Therefore, throughout this survey, we use the term VLM-based OOD detection to refer broadly to OOD detection methods that leverage VLMs, and we similarly use the prefix VLM-based for other tasks (*e.g.*, VLM-based AD). However, it is important to note that not all VLMs employ softmax-based scoring mechanisms—for example, SigLIP (Zhai et al., 2023) uses sigmoid-based classification. As a result, some methods (Ming et al., 2022a; Miyai et al., 2025b), which rely on softmax outputs, cannot be directly applied to such models. For clarity and consistency, this survey focuses primarily on CLIP as the representative VLM and investigates VLM-based methods for OOD detection and related tasks within this scope.

As OOD detection research is primarily focused on the image domain, we conduct a survey of other tasks within the image domain that are common and have strong connections to OOD detection research. For instance, our survey does not cover video domain tasks (Du et al., 2024; Zara et al., 2023; Wu et al., 2024b) due to their limited connection to OOD detection. From the results in Table 1, it is clear that Sensory AD and OOD detection have a very high number of papers, while the other fields are not as active. In the next section, we will examine these findings in greater depth and discuss the evolution of each field.

### 2.3 Generalized OOD Detection v2: Evolution in VLM Era

We propose a generalized OOD detection v2, encapsulating the evolution of each field in the VLM era. The evolution trajectory of the Generalized OOD detection framework v2 is shown in Fig. 2. The evolution of each field is as follows:

**(a) Sensory AD → VLM-based AD**  Sensory AD has continued to develop as a common problem setting for VLM-based AD, inheriting the challenges of traditional sensory AD (Jeong et al., 2023; Deng et al., 2023; Chen et al., 2023c;b; Tamura, 2023; Chen et al., 2023c; Zhou et al., 2024a; Gu et al., 2024a; Li et al., 2024c; Zhu & Pang, 2024). As shown in Table 1, the first appearance in a top venue was at CVPR 2023, and since then, numerous papers have been published in top venues. Therefore, it is evident that Sensory AD has consistently been an active research field.

**(b) Semantic AD/ND → Inactive**  Research on semantic AD/ND appears to become inactive in the VLM era. As shown in Table 1, there are three papers, TMLR 2022 (Liznerski et al., 2022), CVPR 2024 (Zhu & Pang, 2024), and ICLR 2025 (Yun et al., 2025). However, the CVPR 2024 (Zhu & Pang, 2024) and ICLR 2025 (Yun et al., 2025) work aims to build a generalist anomaly detector that solves many AD tasks, including sensory AD and semantic AD, and is not primarily focused on semantic AD. The reasons for the inactivity include saturation of performance for one-class semantic AD/ND, and incompatibility of CLIP and multi-class semantic AD/ND settings. As for one-class semantic AD/ND, TMLR (Liznerski et al., 2022) exists, but the performances with common CIFAR and ImageNet-30 datasets have already achieved around 99%. As for multi-class semantic AD/ND, a common approach is to treat ID classes as a single class, but treating ID classes as a single class is less compatible with CLIP's class-wise discriminative capability.

**(c) OSR → VLM-based OOD Detection**  We consider that OSR has been integrated into VLM-based hard OOD detection. According to Table 1, there is only one top venue publication (Miller et al., 2024) on OSR research in the VLM era. Originally, the main difference between OSR and OOD detection was the benchmark setup (Yang et al., 2024; Salehi et al., 2022). OSR typically divides the classes in the one dataset into some known (ID) classes and unknown (OOD) classes, as seen in MNIST-4/6 (Deng, 2012) CIFAR-4/6 (Krizhevsky et al., 2009a), CIFAR-50/50 (Krizhevsky et al., 2009b), and TinyImageNet-20/180 (Torralba et al., 2008). However, in recent years, some works on VLM-based OOD detection incorporate the benchmark setup of OSR and create new benchmarks such as ImageNet-10/ImageNet-20 (Ming et al., 2022a) and ImageNet-protocol (Palechor et al., 2023; Li et al., 2024b) for hard OOD detection. The only work (Miller

et al., 2024) discusses the OSR problem in the VLM era, but they do not mention the differences from VLM-based OOD detection, and the definitions of VLM-based OSR are identical to those of VLM-based OOD detection. Considering the number of papers on VLM-based OOD detection and its identical definitions with OSR, we consider that the boundary between OOD detection and OSR has effectively disappeared, and all research in the VLM era has been integrated into OOD detection.

Nevertheless, while pure OSR research is declining, some studies have used the term "open-set" in the context of domain generalization (Shu et al., 2023). These studies deviate from the original scope of OSR research and are rather closely aligned with the field of domain generalization (Zhou et al., 2022a). Therefore, within our generalized OOD detection v2, we do not classify these studies as falling under OSR research.

**(d) OOD Detection → VLM-based OOD Detection**   OOD detection is a highly active research area in the VLM era. As shown in Table 1, there are many papers in top venues, indicating a high interest from the community. Additionally, as mentioned above, OSR has been integrated with OOD detection as a field of hard OOD detection (Ming et al., 2022a; Li et al., 2024b). Therefore, it is expected that OOD detection will continue to grow and develop further.

**(e) OD → Inactive**   OD has become less active in the VLM era. Previously, OD was used for open-set semi-supervised learning (Yu et al., 2020; Saito et al., 2021; Cao et al., 2022), learning with open-set noisy labels (Wang et al., 2018), and novelty discovery (Han et al., 2019; Zhao & Han, 2021; Jia et al., 2021; Vaze et al., 2022b; Joseph et al., 2022). The reason for the inactivity is that the use of CLIP led to a reduction in training costs and only a small amount of data needs to be collected, eliminating the need for large amounts of unlabeled data and reducing the need to consider noisy data. However, recently, Liang et al. (2024) proposed Unsupervised Universal Fine-Tuning, a new problem setting for VLM-based OD in ICML2024. Unsupervised Universal Fine-Tuning assumes a more realistic problem setting for unsupervised tuning of the downstream task with CLIP where some OOD samples are included in the unlabeled samples. With this new problem setting, there is still a possibility that OD will become active in the future. However, as OD is not currently an active area, we exclude OD from the main discussion of this survey. Unsupervised Universal Fine-Tuning is deeply related to OOD detection and will be discussed in detail in Sec. 4.3.

## 2.4   Discussion

Through Sec. 2.3, we found that previously mixed fields have been correctly organized in the VLM era, and that the focus becomes OOD detection and sensory AD. These fields are still developing, with an increasing number of methodologies and benchmarks, and are expected to become more active in the future. Note here that this does not mean that other fields have come to an end. For example, one reason why one-class semantic AD/ND has not been studied is the saturation of performance (Liznerski et al., 2022). If more fine-grained and challenging datasets could be constructed, the field could be reactive. We put this in out-of-scope for this survey paper, but this is an important future challenge.

## 3   Overview of Each Problem in VLM Era

In addition to the above inter-field evolution, we emphasize that the advent of VLMs has significantly changed the field of OOD detection itself. In this section, we present an overview of VLM-based OOD detection, highlighting the key changes in the problem definition, the problem setting, and benchmarks. In addition, we also present an overview of VLM-based AD in the hope that the understanding of each field will lead to a deeper understanding of VLM-based OOD detection. We describe the changes in problem definition, problem setting, and benchmarks in the VLM era. As for the background, applications, and evaluation in each field, we refer the readers to the original generalized OOD detection paper (Yang et al., 2024).

### 3.1   VLM-based Out-of-Distribution Detection

**Definition**   The definition of VLM-based OOD detection differs significantly from that of conventional OOD detection. Conventional OOD detection aims to detect test samples drawn from a distribution that is different from the training distribution. As another definition, OOD detection is defined as a task to detect test

samples that the model cannot or does not want to generalize (Yang et al., 2024). However, for VLM-based OOD detection, CLIP has a vast amount of knowledge, so the OOD sample is completely unrelated to the distribution of the CLIP's pretraining data or the CLIP's own generalization ability. Therefore, traditional definitions cannot adequately describe the definition of VLM-based OOD detection.

Unlike the previous definition, VLM-based OOD detection is defined as follows (Ming et al., 2022a; Esmaeilpour et al., 2022): VLM-based OOD detection aims to detect samples that do not belong to any ID class text provided by the user. Given a pre-trained model, a classification task of interest is defined by a set of class labels $\mathcal{Y}_{\text{ID}}$, which we refer to as the ID classes. The semantic distribution is represented by the distribution $P(\mathcal{Y}_{\text{ID}})$. VLM-based OOD detection aims to detect test samples that come from the distribution with the semantic shift from the ID classes, *i.e.*, $P(\mathcal{Y}_{\text{ID}}) \neq P(\mathcal{Y}_{\text{OOD}})$. Following the definition of the generalized OOD detection framework (Yang et al., 2024), ideal OOD detectors should keep the classification performance on test samples from ID class space $\mathcal{Y}_{\text{ID}}$, and reject OOD test samples with semantics outside the support of $\mathcal{Y}_{\text{ID}}$.

**Problem Setting** VLM-based OOD detection focuses on solving the image classification task in a computationally efficient way. Unlike traditional OOD detection settings, which primarily involve training an ID classifier with whole ID data, VLM-based OOD detection primarily focuses on a zero-shot (Ming et al., 2022a) (*i.e.*, without utilizing ID images) or few-shot (Miyai et al., 2023b) (*i.e.*, utilizing only a few ID images) setting. Each detailed definition of both settings is described later in Sec. 4. The field is advancing towards greater computational efficiency, requiring minimal or no training data.

**Benchmark** The benchmark has also changed between VLM-based methods and previous approaches. Earlier works before CLIP often utilized small-scale datasets such as CIFAR (Krizhevsky et al., 2009a;b) and MNIST (Deng, 2012). In contrast, most recent works in VLM-based OOD detection use high-resolution and large-scale datasets such as ImageNet (Ming et al., 2022a; Miyai et al., 2023b; Bai et al., 2024a; Li et al., 2024b; Cao et al., 2024a). The common ImageNet OOD benchmark uses ImageNet as ID and other datasets (Van Horn et al., 2018; Zhou et al., 2017; Xiao et al., 2010; Cimpoi et al., 2014) as OOD. However, in this common benchmark, the semantics between ID and OOD are far, which may allow easy distinction between the ID and OOD. Therefore, recent works use more challenging OOD benchmarks where they split ImageNet classes into ID and OOD categories for hard OOD detection (Ming et al., 2022a; Li et al., 2024b; Jung et al., 2024). The representative datasets are ImageNet-20 (Ming et al., 2022a), ImageNet-10 (Ming et al., 2022a), and ImageNet-protocol (Palechor et al., 2023) created by dividing into multiple variations of ID/OOD pairs from ImageNet-1K. This creation strategy initially focused on OSR but has recently been repurposed for OOD detection. More recently, Noda et al. (2025) proposed ImageNet-X for hard OOD detection and ImageNet-FS-X and Wilds-FS-X for hard full-spectrum OOD detection, which include more challenging conditions reflecting real-world scenarios. These changes in the datasets shift OOD detection closer to the real world and make it a more challenging and practical task.

## 3.2 VLM-based Anomaly Detection

**Definition** Unlike OOD detection, the definition of anomaly detection (AD) has not changed between conventional AD and VLM-based AD. AD is intended for use in specific circumstances (industrial inspection), where samples that deviate from predefined normality are considered an anomaly (Yang et al., 2024; Ruff et al., 2021). Whether a model can generalize is irrelevant to the definition of "Anomaly". Therefore, even with the emergence of CLIP, the definition has not changed.

**Problem Setting** VLM-based AD focuses on solving anomaly classification and segmentation in a computationally efficient way. Anomaly classification is a binary classification task that distinguishes between normality and abnormality. Anomaly segmentation involves segmenting the location of anomalies. Like VLM-based OOD detection, VLM-based AD also primarily focuses on a zero-shot (Jeong et al., 2023) (*i.e.*, without utilizing images in the target dataset) or few-shot (Jeong et al., 2023) (*i.e.*, using only a few images in the target dataset) setting. Each detailed definition of zero-shot and few-shot settings is described later in Sec. 5. As another shift, conventional AD created separate models for each category (Bergmann et al., 2020; Defard et al., 2021; Li et al., 2021; Liznerski et al., 2021; Yi & Yoon, 2020; Zavrtanik et al., 2021; Thulasidasan

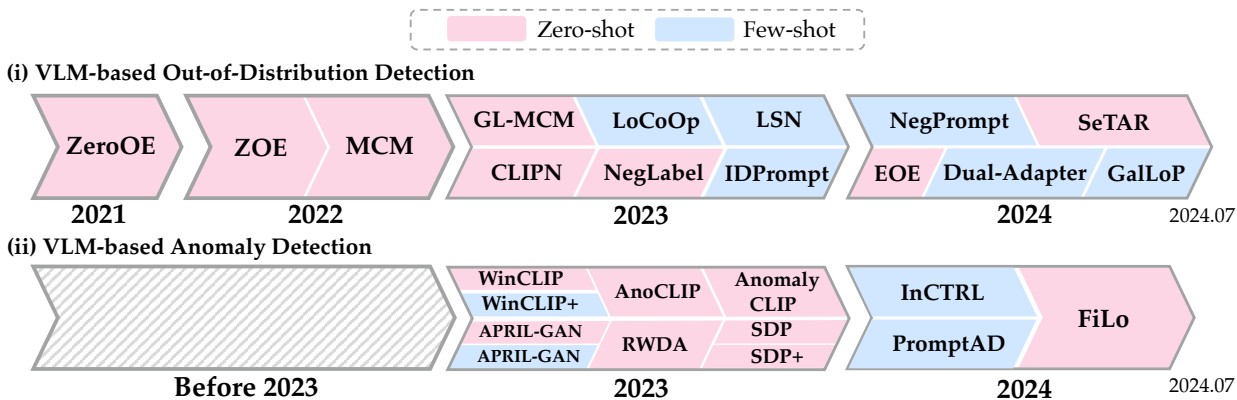

Figure 3: Timeline for representative methodologies for VLM-based out-of-distribution detection and VLM-based anomaly detection. We observe that an increasing number of methods have recently been proposed for both tasks, indicating the growing activity in these fields.

et al., 2021), while VLM-based AD creates a single unified model across multiple categories (Jeong et al., 2023; Deng et al., 2023; Zhou et al., 2024a; Chen et al., 2023b; Zhu & Pang, 2024), which leads to a more computationally efficient approach.

One key difference from VLM-based OOD detection is that VLM-based OOD detection does not involve localization tasks, while these are mainstream in VLM-based AD. This will be discussed in detail in Sec. 5.4.

**Benchmark** Most works on VLM-based AD tackle industrial inspection (Bergmann et al., 2019; Chu & Kitani, 2020; Atha & Jahanshahi, 2018). As for the benchmarks, MVTec-AD dataset (Bergmann et al., 2019) and VisA dataset (Zou et al., 2022) are commonly used. MVTEC-AD includes 15 classes including 5 different texture categories and 10 different object categories with totally 5,354 high-resolution images. VisA covers 12 objects such as in 3 domain consisting of 10,821 high-resolution color images. The VisA benchmark includes objects with complex structures such as printed circuit boards and multiple instances with different locations within a single view, making it one of the most challenging datasets currently available in the open datasets. As for other datasets, KSDD (Tabernik et al., 2020), BTAD (Mishra et al., 2021), and MPDD (Jezek et al., 2021) are often used. However, apart from MVTec-AD and VisA, the datasets used vary depending on the papers (Yang et al., 2025; Cao et al., 2024c; Qu et al., 2024; Li et al., 2024f).

## 4 VLM-based OOD Detection: Methodology

In this section, we introduce the methodologies for VLM-based out-of-distribution (OOD) detection. Fig. 3 presents the timeline for representative methodologies for VLM-based OOD detection. Table 2 presents representative methods. We introduce methods for zero-shot OOD detection in Sec. 4.1, few-shot OOD detection in Sec. 4.2, and other research directions in Sec. 4.3. We categorize each methodology by the training type and the use of OOD prompts. As for the OOD prompts, the methods that employ additional OOD prompts and do not use them are shown in Fig. 4 (a) and (b), respectively. As for the training type, we categorize the methods into training-free, ID training, and auxiliary training (training with other data than ID training data). The illustration of each training scenario is shown in Fig. 5.

### 4.1 Zero-shot Out-of-Distribution Detection

Zero-shot OOD detection was proposed in 2021 by Fort et al. (2021). Since then, a growing number of methods have been proposed year by year.

**Definition of Zero-shot OOD Detection** In zero-shot OOD detection, the term "Zero-shot" refers to the non-use of ID images during both training and inference phases. For instance, the method with additional training with auxiliary datasets (non-use of ID images) can be regarded as a zero-shot method (Wang

Table 2: Representative paper list for VLM-based out-of-distribution detection and anomaly detection.

| Task | ID Image Availability | Training Type | OOD Prompts | Methods |
|---|---|---|---|---|
| § 4 VLM-based OOD Detection | § 4.1 Zero-shot | § 4.1.1 Training-free | ✓ | ZeroOE (Fort et al., 2021), ZOC (Esmaeilpour et al., 2022), NegLabel (Jiang et al., 2024), EOE (Cao et al., 2024a), AdaNeg (Zhang & Zhang, 2024), CSP (Chen et al., 2024a) |
| | | | ✗ | MCM (Ming et al., 2022a), GL-MCM (Miyai et al., 2025b), SeTAR (Li et al., 2024e), TAG (Liu & Christopher, 2024), CoVer (Zhang et al., 2024a) |
| | | § 4.1.2 Auxiliary Training | ✓ | CLIPN (Wang et al., 2023a) |
| | § 4.2 Few-shot | § 4.2.1 ID Training | ✗ | PEFT-MCM (Ming & Li, 2024a), LoCoOp (Miyai et al., 2023b), GalLoP (Lafon et al., 2024), SCT (Yu et al., 2024) |
| | | | ✓ | LSN (Nie et al., 2023), NegPrompt (Li et al., 2024b), ID-like-Prompt (Bai et al., 2024a), Local-Prompt (Zeng et al., 2025) |
| | | § 4.2.2 Training-free | ✗ | Dual-Adapter (Chen et al., 2024b), DPM (Zhang et al., 2024f) |
| § 5 VLM-based Anomaly Detection | § 5.1 Zero-shot | § 5.1.1 Training-free | ✓ | WinCLIP (Jeong et al., 2023), AnoCLIP (Deng et al., 2023), SDP (Chen et al., 2023c), ALFA (Zhu et al., 2024a) |
| | | § 5.1.2 Auxiliary Training | ✓ | APRIL-GAN (zero-shot) (Chen et al., 2023b), RWDA (Tamura, 2023), SDP+ (Chen et al., 2023c), AnomalyCLIP (Zhou et al., 2024a), FiLo (Gu et al., 2024a), AdaCLIP (Cao et al., 2024c), VCP-CLIP (Qu et al., 2024), AA-CLIP (Ma et al., 2025), Bayes-PFL (Qu et al., 2025), |
| | § 5.2 Few-shot | § 5.2.1 Training-free | ✓ | WinCLIP+ (Jeong et al., 2023), UniVAD (Gu et al., 2025) |
| | | § 5.2.2 ID Training | ✓ | CLIP-FSAC (Zuo et al., 2024), PromptAD (one-class) (Li et al., 2024c), One-to-Normal (Li et al., 2024f) |
| | | § 5.2.3 Auxiliary Training + ref. | ✓ | APRIL-GAN (few-shot) (Chen et al., 2023b), InCTRL (Zhu & Pang, 2024) |

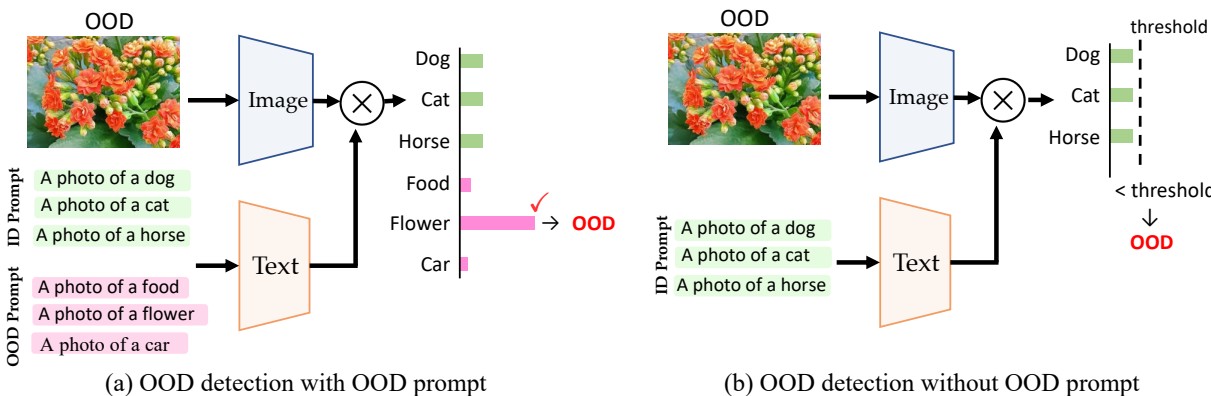

(a) OOD detection with OOD prompt

(b) OOD detection without OOD prompt

Figure 4: Illustration of the OOD detection methods with OOD prompt and without OOD prompt.

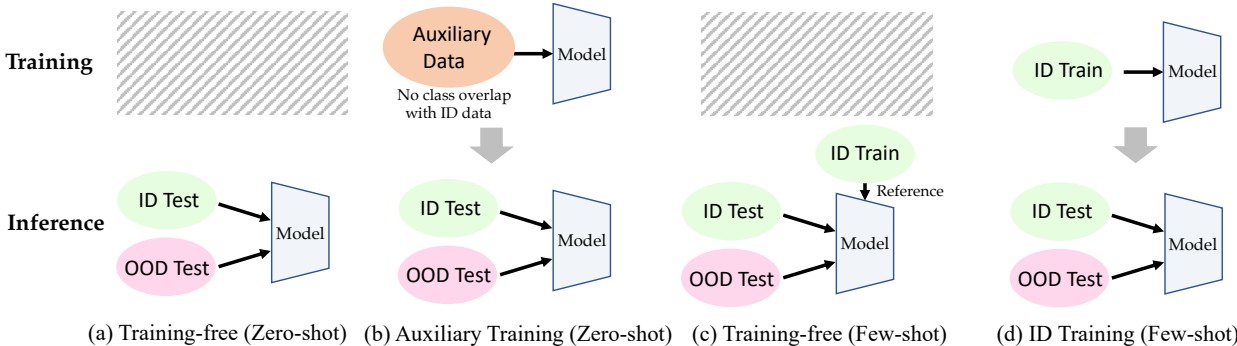

(a) Training-free (Zero-shot)  (b) Auxiliary Training (Zero-shot)  (c) Training-free (Few-shot)  (d) ID Training (Few-shot)

Figure 5: Illustration of the OOD detection methods for each training type in both zero-shot and few-shot settings.

et al., 2023a). The method with the pre-processing of the ID class texts can also be regarded as a zero-shot method (Fort et al., 2021; Esmaeilpour et al., 2022; Jiang et al., 2024; Cao et al., 2024a).

### 4.1.1 Training-free Methods

**a. With OOD Prompts**   VLM-based OOD detection started in this setting. The earliest work is ZeroOE (Fort et al., 2021). ZeroOE feeds the potential OOD labels to the textual encoder of CLIP. However, the method of using known OOD labels is infeasible for real-world applications. To solve this problem, ZOC (Esmaeilpour et al., 2022) proposed to train an OOD label generator based on the visual encoder of CLIP and use the generated pseudo-OOD labels for OOD detection. However, when dealing with large-scale datasets encompassing a multitude of ID classes, the label generator may not generate effective candidate OOD labels, resulting in poor performance. Building on these early works (Fort et al., 2021; Esmaeilpour et al., 2022), recent works focus on how to obtain high-quality OOD labels through either (i) OOD label retrieval (Jiang et al., 2024; Ding & Pang, 2024) or (2) OOD label generation (Cao et al., 2024a). (i) A representative retrieval-based method is NegLabel (Jiang et al., 2024). NegLabel selects high-quality OOD labels from extensive corpus databases by calculating the distance between an extracted OOD label and ID label. AdaNeg (Zhang & Zhang, 2024) extends NegLabel by incorporating adaptive negative proxies, which are dynamically generated during inference by leveraging actual OOD images. This adaptation ensures a closer alignment with the test-time OOD label space, thereby enhancing the effectiveness of negative proxy guidance. Chen et al. (2024a) proposed Conjugated Semantic Pool (CSP), which enhances OOD detection by expanding the semantic space with modified superclass names. (ii) A representative generation-based method is EOE (Cao et al., 2024a). EOE utilizes Large Language Models (LLMs) to produce high-quality OOD labels. By modifying the prompts given to the LLM, EOE can be generalized to a variety of tasks, including far and near OOD detection.

**b. Without OOD Prompts** In zero-shot OOD detection, many methods utilize OOD labels, but the difficulty and cost of creating these labels pose challenges. To address these issues, Ming et al. (2022a) proposed MCM, which uses only ID labels to detect OOD. MCM is a simple approach that devises softmax scaling to align visual features with textual concepts for OOD detection. Despite its simplicity, MCM has high effectiveness and scalability, and it serves as a crucial baseline in VLM-based OOD detection. Building on the concept of MCM, Miyai et al. (2025b) proposed GL-MCM, which extends MCM by just adding a local MCM score to enhance the fine-grained detection capability in local regions. Based on MCM and GL-MCM SeTAR (Li et al., 2024e) enhances them by changing the model's weight matrices using a simple greedy search algorithm. As for another direction, some works adopt data-centric approaches (Liu & Christopher, 2024; Zhang et al., 2024a). Liu & Christopher (2024) proposed TAG, which applies simple augmentations to the ID text prompts. Zhang et al. (2024a) proposed CoVer, which enhances separability by averaging confidence scores across corrupted inputs, leveraging the greater confidence drop in OOD samples under corruptions.

We consider these methods to be post-hoc methods for VLM-based OOD detection in that they directly employ an ID classifier for OOD detection. Due to their simplicity and high scalability, these post-hoc methods can bring fundamental performance improvements for many subsequent methods (Miyai et al., 2023b; Nie et al., 2023; Li et al., 2024b). Therefore, we expect that this field should be developed further in the future, reflecting the trajectory of the field before CLIP emerged (Hendrycks & Gimpel, 2017; Liang et al., 2018; Lee et al., 2018; Liu et al., 2020; Sastry & Oore, 2020; Wang et al., 2021; Zhang et al., 2023; Sun & Li, 2022; Sun et al., 2022; Lin et al., 2021; Sastry & Oore, 2019).

### 4.1.2 Auxiliary Training-based Methods

CLIPN (Wang et al., 2023a) is the only auxiliary training-based method for zero-shot OOD detection. CLIPN aims to empower the logic of saying "no" within CLIP, and it designs a novel learnable "no" prompt and an additional "no" text encoder to capture negation semantics within images. To create an additional text encoder, CLIPN needs to be pre-trained on the CC-3M dataset (Sharma et al., 2018). While the extensive pre-training of CLIPN may indeed lead to intensive computations and lower scalability, once pre-trained, it performs zero-shot OOD detection across a wide range of domains with comparable performance to few-shot OOD detection methods (Miyai et al., 2023b; Nie et al., 2023).

### 4.2 Few-shot Out-of-Distribution Detection

Few-shot OOD detection was concurrently proposed by Miyai et al. (2023b) and Ming & Li (2024a) in June 2023. Since then, it has become an active research area in VLM-based OOD detection.

**Definition of Few-shot OOD Detection** VLM-based few-shot OOD detection aims to detect OOD images using only a few labeled ID images. In few-shot OOD detection, the term "Few-shot" refers to the use of a few ID images during training or inference phases. For instance, the method with additional training with a few ID images can be regarded as a few-shot method (Li et al., 2024c). Even without training, if a method uses a few ID images as a reference, it is treated as a few-shot method (Jeong et al., 2023). Regarding the number of shots, it is common to experiment with 1-shot to 16-shot (Miyai et al., 2023b; Chen et al., 2024b), following the closed-set setting (Zhou et al., 2022c).

### 4.2.1 ID Training-based Methods

**a. Without OOD Prompts** Few-shot OOD detection began in this setting. Ming & Li (2024a) proposed PEFT-MCM for VLM-based OOD detection, which demonstrates the effectiveness of combining parameter-efficient tuning methods (*e.g.*, prompt learning (Zhou et al., 2022c) or adapter (Zhang et al., 2022)) and MCM (Ming et al., 2022a). Concurrently, Miyai et al. (2023b) proposed LoCoOp, a pioneer prompt learning approach for few-shot OOD detection. LoCoOp enhances CoOp's (Zhou et al., 2022c) OOD detection capabilities by performing OOD regularization with local OOD features. LoCoOp is the simplest prompt learning method and serves as a crucial baseline in few-shot OOD detection. To improve the performance of LoCoOp, Self-Calibrated Tuning (SCT) (Yu et al., 2024) dynamically adjusts the weight of OOD regularization based on prediction uncertainty to prevent unreliable OOD features from degrading

the model's performance. Unlike LoCoOp, which utilizes non-ID local regions for OOD regularization, GalLoP (Lafon et al., 2024) proposes an approach that utilizes local ID regions to enable a more fine-grained distinction between ID and OOD samples. GalLoP learns a diverse set of prompts by utilizing both global and local visual representations, thereby enhancing detection capabilities.

**b. With OOD Prompts** Similar to zero-shot OOD detection, recent works in few-shot OOD detection utilize additional OOD prompts (Nie et al., 2023; Bai et al., 2024a; Li et al., 2024b). As representative methods, LSN (Nie et al., 2023) and NegPrompt (Li et al., 2024b) were proposed concurrently. They state that the simple negative prompts added "not" (*e.g.*, "not a photo of a [cls]") fail to capture the dissimilarity for identifying OOD samples. Therefore, by preparing negative prompts and training with them, LSN and NegPrompt can learn suitable negative prompts, enabling more accurate detection of OOD samples. The difference between LSN and NegPrompt lies in their approach to the use of negative prompts. LSN prepares unique negative prompts for each class and learns suitable negative prompts for each class. In contrast, NegPrompt prepares multiple negative prompts common to all ID classes and trains them to learn generic templates representing the negative semantics of any given class labels. Additionally, NegPrompt tested the performances in the hard OOD detection setting with ImageNet-protocol (Palechor et al., 2023), outperforming LoCoOp and CoOp. Alternatively, ID-like-Prompt (Bai et al., 2024a) takes a different approach by introducing ID-like prompts, which are designed for capturing OOD features that are close to the ID features. It extracts ID-like OOD regions from ID training images and trains ID-like prompts using these extracted OOD data, enabling the model to capture more subtle differences within images. In a unique direction, LAPT (Zhang et al., 2024d) proposes an automatic sample collection strategy that retrieves or generates training ID images only with ID class names, which achieves high performance without image collection and annotation costs. LAPT then performs distribution-aware prompt learning, which distinguishes between ID class and OOD class tokens. LAPT is positioned within the context of more efficient few-shot OOD detection in this survey paper since it requires generating or retrieving "ID images" for the data collection.

In the context of few-shot OOD detection, recently, Li et al. (2024b) proposed a new problem setting called open-vocabulary OOD (OV-OOD) detection. While common few-shot OOD detection involves training on images from all ID classes during training, OV-OOD detection involves training on images from just a small subset of ID classes and performing OOD detection using all ID classes at inference time. Formally, we define a subset of semantic labels $\mathcal{Y}_{\text{ID,sub}} \subset \mathcal{Y}_{\text{ID}}$, where $\mathcal{Y}_{\text{ID}}$ represents all ID labels. Based on this subset of labels, we define a corresponding subset dataset $\mathcal{D}_{\text{ID,sub}}^{\text{train}} \subset \mathcal{D}_{\text{ID}}^{\text{train}}$. During training, only $\mathcal{D}_{\text{ID,sub}}^{\text{train}}$ is used. Then, at inference time, all ID classes $\mathcal{Y}_{\text{ID}}$ are used, and the goal is to detect OOD from a combination of all ID test data $\mathcal{D}_{\text{ID}}^{\text{test}}$ with $\mathcal{Y}_{\text{ID}}$ and OOD test data $\mathcal{D}_{\text{OOD}}^{\text{test}}$ with $\mathcal{Y}_{\text{OOD}}$. For this setting, existing few-shot OOD detection methods (Miyai et al., 2023b; Li et al., 2024b) can be easily applied by simply combining the rest of the ID classes. In particular, NegPrompt (Li et al., 2024b) learns general negative prompts that are not specific to the training ID classes, so it achieves high performance in OV-OOD detection.

### 4.2.2 Training-free Methods

Recently, training-free few-shot OOD detection methods, which use ID images solely as references during inference, have gained attention. Dual-Adapter (Chen et al., 2024b) adopts a prior-based method Tip-Adapter (Zhang et al., 2022), which leverages both textual and visual features with a cache model and enhances performance without training. To adapt this to few-shot OOD detection, Dual-Adapter employs the concept of dual cache modeling and constructs Positive-Adapter and Negative-Adapter, and identifies OOD samples with the prediction difference with both adapters. Dual-Pattern Matching (DPM) (Zhang et al., 2024f) enhances CLIP for OOD detection by utilizing both textual and visual ID patterns. DPM stores ID class-wise text features as the textual pattern and the aggregated ID visual information as the visual pattern. During inference, it evaluates the similarity of inputs to both patterns to identify OOD samples.

### 4.3 Other Directions

#### 4.3.1 VLM-based Full-spectrum OOD Detection

VLM-based full-spectrum OOD (FS-OOD) detection is a crucial challenge (Lu et al., 2023). FS-OOD detection was proposed by Yang et al. (2023) as an important setting that considers both the detection of semantic shift and the ability to recognize covariate-shifted data (Hendrycks & Dietterich, 2019; Hendrycks et al., 2021a). Unlike standard OOD detection, which only focuses on semantic shifts between training and test distributions, FS-OOD detection further considers non-semantic covariate shift by including covariate-shifted ID images. As for the benchmarks, OpenOOD v1.5 (Zhang et al., 2024b) provides two large-scale benchmarks based on ImageNet-200 and ImageNet-1K, incorporating ImageNet-C (Hendrycks & Dietterich, 2019) with image corruptions, ImageNet-R (Hendrycks et al., 2021a) with style changes, and ImageNet-V2 (Recht et al., 2019) with resampling bias as ID. As for VLM-based methods, LSA (Lu et al., 2023) uses a bidirectional prompt customization mechanism, which adjusts discriminative ID and OOD boundary. The latest work, Noda et al. (2025) introduces ImageNet-FS-X and Wilds-FS-X as new benchmarks. ImageNet-FS-X utilizes the hierarchical structure of ImageNet labels to define a more challenging ID/OOD split while also evaluating robustness to covariate shifts in ImageNet variants. To further align these benchmarks with real-world testbeds, Wilds-FS-X extends Wilds (Koh et al., 2021) for FS-OOD evaluation. These benchmarks present substantial room for further improvements.

#### 4.3.2 Other Tasks with VLM-based OOD Detection

**Unsupervised Universal Fine-Tuning**   VLM-based OOD detection is useful for a new task called Unsupervised Universal Fine-Tuning (UUFT) (Liang et al., 2024). UUFT is a problem of unsupervised learning for outlier detection (OD). Existing studies for unsupervised learning assumed that all unlabeled images belong to one of the ID classes (Huang et al., 2022; Shu et al., 2022; Tanwisuth et al., 2023), but they require prior knowledge of exact class names linked to ground truth labels, which restricts their usefulness in various real-world situations. For a more realistic setting, UUFT assumes that OOD images are included in the unlabeled images. To detect OOD images during training, they developed MCM (Ming et al., 2022a) and proposed UEO, which leverages sample-level confidence to approximately minimize the conditional entropy of confident instances and maximize the marginal entropy of less confident instances.

**Open-world Prompt Tuning**   VLM-based OOD detection is useful for a new task called Open-world Prompt Tuning (Zhou et al., 2024b). Open-world Prompt Tuning is a task that evaluates the classification accuracy on a mix of known and novel ID classes while training the model with known classes. To solve this problem, Zhou et al. (2024b) proposed DeCoOp which incorporates OOD detection into the inference pipelines and improves the base-to-new separability, preventing performance degradation on new classes.

## 5 VLM-based AD: Methodology

In this section, we introduce methodologies for VLM-based anomaly detection (AD) in the hope that the contrast with OOD detection clarifies the similarities and differences between each task and facilitates a deeper understanding of VLM-based OOD detection.

**Common Findings in All Settings**   In VLM-based AD, all representative methods utilize anomaly prompts (*e.g.*, "`anomalous [class]`") to detect anomalies. This hypothesis is supported by several observations from existing work (Jeong et al., 2023). Firstly, the concepts of normality and anomalies are context-dependent states (Isola et al., 2015) of an object, with language playing a crucial role in defining these states. Secondly, language provides additional insights that help differentiate defects from acceptable variations in normality.

### 5.1 Zero-shot Anomaly Detection

VLM-based zero-shot AD was proposed in 2023 by Jeong et al. (2023). Although it started about two years later than OOD detection, many methods have been proposed up to the present.

**Definition of Zero-shot AD**    The meaning of the term "Zero-shot" for zero-shot AD is similar to that for zero-shot OOD detection. In zero-shot AD, the term "Zero-shot" refers to the non-use of the images in the target domain during both training and inference phases. For instance, the method with additional training with auxiliary datasets can be regarded as a zero-shot method (Chen et al., 2023b; Tamura, 2023; Chen et al., 2023c; Zhou et al., 2024a; Gu et al., 2024a). The method with the pre-processing of the target class texts can also be regarded as a zero-shot method (Jeong et al., 2023; Deng et al., 2023; Chen et al., 2023c).

### 5.1.1   Training-free Methods

The simplest zero-shot AD methods are (i) to perform anomaly classification with CLIP using text prompts for normality and anomalies as classes (*i.e.*, "`normal [class]`" *vs.* "`anomalous [class]`") and (ii) to calculate the similarity to the normal prompt (*i.e.*, "`normal [class]`") as the score.  These methods are called CLIP-AC (Jeong et al., 2023). Jeong et al. (2023) reported that CLIP-AC with both normal and anomaly prompts outperforms that with only normal text prompts, which indicates the importance of the use of anomaly prompts. However, the performances for this naive method are not yet satisfactory due to the wide range of variations of anomalies. To solve this issue, Jeong et al. (2023) proposed WinCLIP. WinCLIP performs a compositional ensemble on a large number of pre-defined normal and anomaly templates and efficient extraction and aggregation of window/patch/image-level features aligned with the text. WinCLIP outperforms CLIP-AC by a large margin.  Because of its simplicity and pioneering work, WinCLIP has become an important baseline for VLM-based AD. AnoCLIP (Deng et al., 2023) follows WinCLIP's approach of using a large number of pre-defined normal and anomaly templates but modifies the templates to be domain-aware (*e.g.*, industrial photo) and contrastive state for normal and anomaly (*e.g.*, perfect and imperfect). However, it is noteworthy that the performance of the ensemble strategies of previous methods heavily depends on the text descriptions (Jeong et al., 2023; Deng et al., 2023).  Also, it is observed that more descriptions are not always better (Chen et al., 2023c), which makes the previous approaches (Jeong et al., 2023; Deng et al., 2023) using a naive ensemble of large templates somewhat uncontrollable and random in their applications. Therefore, SDP (Chen et al., 2023c) proposes RVS, a representative vector selection paradigm, which makes the mechanism of extracting representative vectors from large templates controllable, allowing for a more diverse selection of representative vectors. As a more recent work, ALFA is a zero-shot anomaly detection framework that leverages LLMs to generate adaptive anomaly descriptions at runtime and optimizes the image-text alignment of CLIP at both global and local levels, enhancing detection accuracy and interpretability (Zhu et al., 2024a).

### 5.1.2   Auxiliary Training-based Methods

Existing methods with auxiliary training perform training on the test set of one dataset and perform zero-shot testing on the other dataset (Chen et al., 2023b; Zhou et al., 2024a; Gu et al., 2024a) (*e.g.*, training with MVTec-AD and evaluation with VisA.) In recent years, the development of auxiliary training-based methods has received more attention than training-free zero-shot methods. There are two main reasons why training is necessary for AD: (i) The first is the domain gap between semantics and anomalies. CLIP is pre-trained to understand the semantics of images, so when applied in a zero-shot manner, it captures the semantics of the image. However, actual anomalies are not semantics, but rather represent the state of an object and appear only in local areas of the image. Therefore, without training, this domain gap between semantics and anomalies cannot be bridged. (ii) The second reason is that there are limitations to relying on a large set of manually crafted anomaly prompts. This incurs prompt creation costs and also makes it difficult to respond to unknown anomalies.

To address the above issue (i), APRIL-GAN (also known as VAND) (Chen et al., 2023b) was proposed. APRIL-GAN tackles the domain gap between semantics and anomaly by adding additional linear layers in vision encoders. These linear layers project image features at each scale into the text space, creating and aggregating anomaly maps at each stage. Similarly, SDP+ (Chen et al., 2024b) also incorporates additional linear layers into SDP (Chen et al., 2024b) to effectively project image features into the text feature space, addressing the misalignment between image and text. To solve both the issue (i) and (ii), AnomalyCLIP (Zhou et al., 2024a) was proposed. AnomalyCLIP is a textual prompt learning-based method similar to CoOp. By replacing anomaly prompts with learnable parameters, it eliminates the need to prepare a large number

of manually pre-defined prompts such as those in WinCLIP (Jeong et al., 2023). Furthermore, unlike CoOp (Zhou et al., 2022c) which learns object semantics, AnomalyCLIP learns object-agnostic text prompts that capture generic normality and abnormality in an image regardless of its semantics. To achieve this, AnomalyCLIP introduces object-agnostic text prompt templates for both normal and anomaly and performs global and local context optimization. FiLo (Gu et al., 2024a) leverages Large Language Models (LLMs) to generate fine-grained anomaly descriptions for each object category. This method replaces generic abnormal descriptions with LLM-generated specific anomaly content for each sample. By adding learnable prompts before the generated anomaly prompts, FiLo performs global and local context optimization, enhancing the ability to detect anomalies. As a more recent method, AdaCLIP (Cao et al., 2024c) employs static and dynamic learnable prompts in both the image and text encoders. Static prompts provide a general adaptation for zero-shot detection, while dynamic prompts are generated per test image for adaptive refinement. Their combination, termed hybrid prompts, enhances zero-shot performance.

As a unique direction from these methods, RWDA (Tamura, 2023) proposes a data augmentation approach by utilizing CLIP's text embeddings as training data. RWDA adds randomly generated words into normal and anomaly prompts to generate a diverse set of normal and anomaly training samples and trains a regular feed-forward neural network with diverse text embeddings. VCP-CLIP (Qu et al., 2024) proposes to leverage visual context prompting (VCP) to enhance CLIP's ability to detect and segment anomalies without requiring product-specific text prompts. VCP eliminates reliance on predefined textual descriptions and improves generalization across unseen product categories.

## 5.2 Few-shot Anomaly Detection

VLM-based few-shot AD was proposed in 2023 by Jeong et al. (2023), concurrently with the development of zero-shot AD (Jeong et al., 2023).

**Definition of Few-shot AD**   VLM-based few-shot AD aims to detect anomaly images using only a few images in the target domain. The meaning of the term "Few-shot" is similar to that of few-shot OOD detection. In few-shot AD, the term "Few-shot" refers to the use of a few images in the target domain during training or inference phases. For instance, the method with additional training with a few images in the target domain can be regarded as a few-shot method (Li et al., 2024c). Even without training, if a method uses a few images in the target domain as a reference, it is regarded as a few-shot method (Jeong et al., 2023).

### 5.2.1 Training-free Methods

The earliest approach in VLM-based few-shot AD is WinCLIP+ (Jeong et al., 2023), an improved method of WinCLIP. WinCLIP, a base zero-shot AD method, cannot identify certain defects that can only be defined visually rather than textually. For example, the "Metal-nut" category in MVTecAD has an anomaly type labeled "flipped upside down," which can only be identified relative to a normal image. To address this, WinCLIP+ incorporates a few normal reference images into a memory bank (Roth et al., 2022) and calculates the anomaly score with the cosine similarity between the query image and its most similar image in the memory bank.

### 5.2.2 ID Training-based Methods

Training-based methods for VLM-based AD are generally categorized as auxiliary training (in Sec. 5.1.2), but some approaches use data from the same domain as the test data for training. CLIP-FSAC (Zuo et al., 2024) enhances few-shot anomaly classification by introducing a two-stage fine-tuning framework with modality-specific adapters, where the image adapter is trained first using text features, followed by the text adapter while freezing the image adapter. To address data scarcity, it employs synthetic anomaly generation using random perturbation and Poisson-based editing to create diverse training samples. PromptAD (Li et al., 2024c) proposes a prompt learning method for one-class AD (where the normal class consists of one class). In one-class AD, traditional prompt learning methods for multi-class classification (*e.g.*, CoOp (Zhou et al., 2022c)) do not work well. To address this, PromptAD creates a large number of anomaly prompts by adding a learnable anomaly suffix to the normal prompt. It then learns to bring the visual features closer

to the normal prompt and further away from the anomaly prompts. As a different direction, Kwak et al. (2024) proposed ADP, which learns anomaly-aware prompts using the personalization capabilities of diffusion models. Also, Li et al. (2024f) has proposed a novel anomaly personalization approach, which applies a one-to-normal transformation to query images using a customized anomaly-free generation model, ensuring alignment with the normal data manifold. This method leverages a triplet contrastive anomaly inference strategy, incorporating comparisons between the query image, a generated anomaly-free data pool, and textual prompts to enhance prediction stability and robustness.

### 5.2.3 Auxiliary Training- and Reference-based Methods

We explore the methods trained on auxiliary datasets and utilize the normal images in the target domain as references during inference. An early work in this category is APRIL-GAN (few-shot) (Chen et al., 2023b), which uses a linear layer trained with auxiliary datasets. Similar to WinCLIP+ (Jeong et al., 2023), APRIL-GAN (few-shot) utilizes a few ID reference images with a memory bank-based approach (Roth et al., 2022). More recently, Zhu & Pang (2024) proposed an in-context-learning-based method called InCTRL. InCTRL trains a model to discriminate anomalies from normal samples by learning to identify residuals or discrepancies between query images and a set of few-shot normal images (in-context sample prompts) from auxiliary data. During inference, InCTRL identifies anomalies by measuring the discrepancy between the features of the query image and a few in-context normal samples from the target dataset.

## 5.3 Other Research Direction

### 5.3.1 Anomaly Detection with Localization Models

Some works (Cao et al., 2023a; Li et al., 2024a) tackle AD using foundation models for localization, such as SAM (Kirillov et al., 2023) or GroundingDINO (Liu et al., 2024e). SAA (Cao et al., 2023a) is a pioneering approach that integrates SAM into anomaly segmentation tasks. It first employs prompt-driven object detection techniques such as GroundingDINO (Liu et al., 2024e) to generate box-level regions conditioned on prompts, highlighting the target anomaly areas. These generated boxes serve as prompts for SAM, which then produces the final anomaly segmentation results. Building on this, SAA+ (Cao et al., 2023a) introduces hybrid prompts that blend domain-specific knowledge with contextual image information, helping to reduce ambiguities inherent in language-based prompts. In contrast, CLIPSAM (Li et al., 2025) replaces GroundingDINO with CLIP, leveraging its superior localization capability to provide more precise prompts for SAM. To bridge the domain gap from natural images, Yang et al. (2025) propose a promptable anomaly segmentation model with SAM, incorporating a novel Self-Perception Tuning (SPT) method. They first design a Self-Draft Tuning strategy, in which SAM initially generates a coarse draft of the anomaly mask, followed by a mask refinement process. Furthermore, they introduce a Visual-Relation-Aware Adapter to enhance the internal perception knowledge of discriminative relation information during decoding.

Given that AD necessitates localization, it is expected that the number of works employing localization foundation models such as SAM will continue to increase.

### 5.3.2 Medical Anomaly Detection

While most works on VLM-based AD focus on industrial AD, recent studies have begun to challenge medical anomaly detection (medical AD) (Chen et al., 2023c; Zhu & Pang, 2024; Huang et al., 2024; Hua et al., 2024; Zhang et al., 2024c; Cao et al., 2024c). VLM-based medical AD is a challenging area as well as industrial AD due to the larger gap between different data modalities. A representative work on medical AD is MVFA (Huang et al., 2024). MVFA is a method specifically tailored for medical AD. It incorporates multiple residual adapters into the CLIP's visual encoder to reduce the domain gap, enabling a stepwise enhancement of visual features across different levels. The future progression of medical AD and industrial AD offers an intriguing perspective, exploring whether these fields will develop independently or influence each other. However, when considering practical applications, it should be noted that medical AD faces the challenge that anomalies are not always describable in the language. Therefore, the development of medical AD methods that do not use CLIP is also important.

### 5.3.3 Video Anomaly Detection

Compared to OOD detection, one important point is that anomaly detection has been actively studied not only in image domains but also in video domains (Zhu et al., 2024b). Video Anomaly Detection (VAD) aims to automatically detect unusual events in videos, and it has many applications such as surveillance and monitoring (Wang et al., 2019b). VAD has several challenges. First, there is no clear and unified definition of anomalies, because what is normal or abnormal (for example, walking on a sidewalk vs. on a highway) depends on the context. Second, anomalies happen rarely and unpredictably, so it is difficult to collect high-quality datasets. This makes it hard to learn such patterns effectively. Because of these challenges and interests, VAD has become an active field, and many survey papers have been published (Suarez & Naval Jr, 2020; Abbas & Al-Ani, 2022; Jiao et al., 2023; Zhu et al., 2024b; Abdalla et al., 2024).

Before the VLM era, most VAD research focused on building task-specific models using unsupervised learning, one-class learning, or weakly supervised learning, because there was a lack of labeled anomaly data (Wu et al., 2024a). After the introduction of VLMs, the research direction changed a lot. Using powerful features from pre-trained VLMs, a lot of work tried zero-shot and few-shot VAD with much less task-specific data (Zanella et al., 2024b). Furthermore, the language understanding capabilities of VLMs have enabled new task settings, such as the integration of textual information (Zanella et al., 2024a;b) and the generation of natural language explanations for detected anomalies (Ye et al., 2025; Zhang et al., 2025a).

These recent trends show that the research direction of VAD has changed a lot with VLMs. On the other hand, OOD detection in video domains is still not widely studied. In this survey, our main goal is to better understand OOD detection itself, while referring to related areas such as AD only as supplementary information. Therefore, we do not give a full review of VAD in this survey, and instead refer readers to existing VAD survey papers (Suarez & Naval Jr, 2020; Abbas & Al-Ani, 2022; Jiao et al., 2023; Zhu et al., 2024b; Abdalla et al., 2024).

### 5.4 Discussion

We discuss the similarities and differences between VLM-based OOD detection and VLM-based AD to deepen our understanding of VLM-based OOD detection.

### 5.4.1 Difference between Each Methodology

**Differing Scopes of OOD** OOD detection and AD differ significantly in the scope of OOD (anomaly) they cover, which leads to differences in methodologies, particularly in the use of OOD prompts. As explained in Sec. 3, sensory AD is intended for specific use cases like industrial inspection, where samples deviating from predefined normality (*e.g.*, defective products) are considered anomalies (Yang et al., 2024; Ruff et al., 2021). In other words, in sensory AD, the anomaly space is limited to damaged objects with shared semantics, and anomalies like images of dogs are not expected. This limited anomaly space allows even simple prompts to achieve decent performance. Therefore, as shown in Table 2, all AD methods utilize anomaly prompts. Conversely, in OOD detection, as explained in Sec. 3.1, anything semantically different from the ID class is considered OOD. Thus, utilizing naive manual OOD prompts is forbidden (even if it improves benchmark performance). This vastness of the OOD space is the key factor differentiating the methodologies between the two fields.

**Difficulty of Localization Task** VLM-based OOD detection and VLM-based AD differ significantly in the difficulty of OOD (anomaly) localization tasks. In VLM-based AD, anomaly segmentation is a mainstream task, performed alongside classification in many papers. However, in VLM-based OOD detection, there has been no research on object-level OOD detection/segmentation. Object-level OOD detection aims to detect OOD objects (Du et al., 2022c;b;a). The inactivity is related to the size of the OOD space, and the too-vast space of OOD makes it difficult to identify OOD objects with prompts effectively. To pave the way for future development, the foundation models for localization such as SAM (Kirillov et al., 2023), which can segment individual objects, have the potential to address object-level OOD detection/segmentation. Object-level OOD detection/segmentation using SAM is a promising future research direction.

### 5.4.2 Similarity between Each Methodology

**Each Problem Setting** The existing problem settings for VLM-based AD and VLM-based OOD detection are similar. Both primarily focus on zero-shot and few-shot settings and can be categorized into training-free, auxiliary training-based, and ID training-based methods. Examining each problem more closely provides valuable insights into future directions for both fields.

**History of Approaches** The history of the progress of methods for VLM-based AD and VLM-based OOD detection is similar. For instance, both problems initially started with naive methods with manual OOD prompts (ZeroOE (Fort et al., 2021) for OOD detection, WinCLIP (Jeong et al., 2023) for AD). To address the issues with these initial approaches, subsequent methods emerged that replaced OOD prompts with learnable parameters (LSN (Nie et al., 2023) and NegPrompt (Li et al., 2024b) for OOD detection, and AnomalyCLIP (Zhou et al., 2024a) for AD). Therefore, by carefully examining each other's fields, there is potential for mutual enhancement and interaction in the future

## 6 Benchmarks and Experiments

In this section, we compare several representative VLM-based OOD detection methods.

### 6.1 Benchmarks and Evaluation Metrics

In OOD detection, it is common to consider an entire dataset as ID and to use several other datasets that are semantically disjoint from any ID classes as OOD datasets. However, as described in Sec.3.1, recent advances in hard OOD detection focus on constructing benchmarks where both ID and OOD samples are derived from the same large-scale dataset such as ImageNet. This setup is inspired by the benchmark design of OSR.

Following prior work (Zhang et al., 2024b; Ming et al., 2022a), we adopt the following terminology: among OOD samples drawn from datasets different from the ID dataset, those that are semantically similar to the ID classes are referred to as near OOD, while those that are semantically distant are referred to as far OOD. When both ID and OOD samples are derived from the same dataset, as in OSR, we refer to this setting as hard OOD.

In this section, we report results on the widely used ImageNet OOD benchmark (Huang & Li, 2021) and two ImageNet-based hard OOD benchmark.

**ImageNet Far OOD Benchmark** In the ImageNet OOD benchmark, ImageNet is used as the ID dataset, while datasets such as iNaturalist (Van Horn et al., 2018) serve as OOD datasets. Among these, the most commonly used benchmark involves using the ImageNet validation set as the ID data and the following four datasets as OOD data: iNaturalist (Van Horn et al., 2018), SUN (Xiao et al., 2010), Places (Zhou et al., 2017), and TEXTURE (Cimpoi et al., 2014). iNaturalist contains approximately 859,000 images of plants and animals across over 5,000 species. For OOD detection evaluation, 10,000 images are randomly sampled from 110 classes that are disjoint from ImageNet-1K. SUN consists of over 130,000 scene images spanning 397 categories. For OOD evaluation, 10,000 images are randomly sampled from 50 categories that do not overlap with ImageNet. Places is another scene-centric dataset with a similar concept space to SUN. For OOD detection, 10,000 images are randomly sampled from 50 non-overlapping classes with ImageNet-1K. TEXTURE comprises 5,640 real-world texture images from 47 categories. The entire dataset is used for OOD evaluation. When training is required, the training set of ImageNet is used for model training.

**ImageNet Near OOD Benchmark** In previous studies, two datasets, SSB-Hard (Vaze et al., 2022a) and NINCO (Bitterwolf et al., 2023), have been proposed as OOD datasets that are more semantically similar to ImageNet. SSB-Hard contains 49,000 images from 980 categories, which are selected from ImageNet-21K (Ridnik et al., 2021). NINCO is a dataset with 5,879 manually collected images (Bitterwolf et al., 2023), which is constructed to be semantically close to ImageNet-1K, but without any overlapping classes.

**ImageNet-20 OOD Benchmark** The ImageNet-20 OOD Benchmark is one of the earliest benchmarks proposed for hard OOD detection (Ming et al., 2022a). In this setting, ImageNet-20 is used as the ID dataset, and ImageNet-10, which has no overlapping categories, is used as the OOD dataset. The direction can also

Table 3: Comparison of OOD detection methods across ImageNet, ImageNet-20, and ImageNet-X. We use AUROC for the evaluation of OOD detection.

| Method | Train Type | OOD Prompts | ImageNet | | | ImageNet-20 | | ImageNet-X | |
|---|---|---|---|---|---|---|---|---|---|
| | | | Far-OOD | Near-OOD | ID Acc. | Hard-OOD | ID Acc. | Hard-OOD | ID Acc. |
| *Zero-shot* | | | | | | | | | |
| MCM | Free | ✗ | 90.66 | 68.91 | 66.73 | 97.38 | 91.30 | 74.30 | 75.97 |
| GL-MCM | Free | ✗ | 91.47 | 71.21 | 66.73 | 98.49 | 91.30 | 75.53 | 75.97 |
| NegLabel | Free | ✓ | **94.15** | 73.33 | 66.73 | 97.47 | 91.30 | 72.15 | 75.97 |
| EOE | Free | ✓ | 92.95 | 70.22 | 66.73 | 98.16 | 91.30 | 77.07 | 75.97 |
| CLIPN-A | Auxiliary | ✓ | 94.11 | **80.94** | 66.73 | 97.16 | 91.30 | **77.08** | 75.97 |
| *Few-shot* | | | | | | | | | |
| CoOp | ID | ✗ | 91.16 | 76.46 | **71.77** | 97.78 | **94.80** | 75.23 | 80.38 |
| LoCoOp | ID | ✗ | 93.02 | 69.11 | 71.61 | **99.04** | 94.67 | 75.17 | **80.42** |
| ID-like-Prompt | ID | ✓ | 92.46 | 67.18 | 68.43 | 92.90 | 89.80 | 66.01 | 77.18 |

be reversed. These subsets are selected with reference to CIFAR-10 (Krizhevsky et al., 2009a) classes, making the semantic distance between ImageNet-20 and ImageNet-10 relatively small. ImageNet-20 contains 1,000 images, while ImageNet-10 contains 500 images.

**ImageNet-X OOD Benchmark**    ImageNet-X is a recently proposed benchmark for hard OOD detection (Noda et al., 2025). It leverages the semantic hierarchy of ImageNet and creates ID and OOD splits within the same superclass. Both ID and OOD sets consist of 500 classes each. The ID and OOD subsets of ImageNet-X contain 25,000 images respectively.

## 6.2 Metrics

We use the ID accuracy and AUROC scores. AUROC scores measure the area under the Receiver Operating Characteristic (ROC) curve.

## 6.3 Experimental Setup

We compare commonly used eight OOD detection methods in zero-shot and few-shot settings.

**Zero-shot Methods**    For the zero-shot setting, we use five methods: MCM (Ming et al., 2022a), GL-MCM (Miyai et al., 2025b), NegLabel (Jiang et al., 2024), EOE (Cao et al., 2024a), and CLIPN (Wang et al., 2023a). As shown in Table 2, MCM and GL-MCM are categorized as methods without training and without OOD prompts. NegLabel and EOE are categorized as methods without training but with OOD prompts. CLIPN is categorized as a method with training and with OOD prompts.

**Few-shot Methods**    For the few-shot settings, we use CoOp (Zhou et al., 2022c; Ming & Li, 2024a), LoCoOp (Miyai et al., 2023b), and ID-like-Prompt (IDPrompt) (Bai et al., 2024a). CoOp and LoCoOp use MCM as the detection method during inference. For CoOp and LoCoOp, we follow the hyperparameter settings from previous studies and train with 16 shots. On the other hand, since IDPrompt requires higher training costs, we follow the original implementation (Bai et al., 2024a) and conduct training with only 1 shot.

## 6.4 Experimental Results

The experimental results are summarized in Table 3. Below, we highlight several key findings.

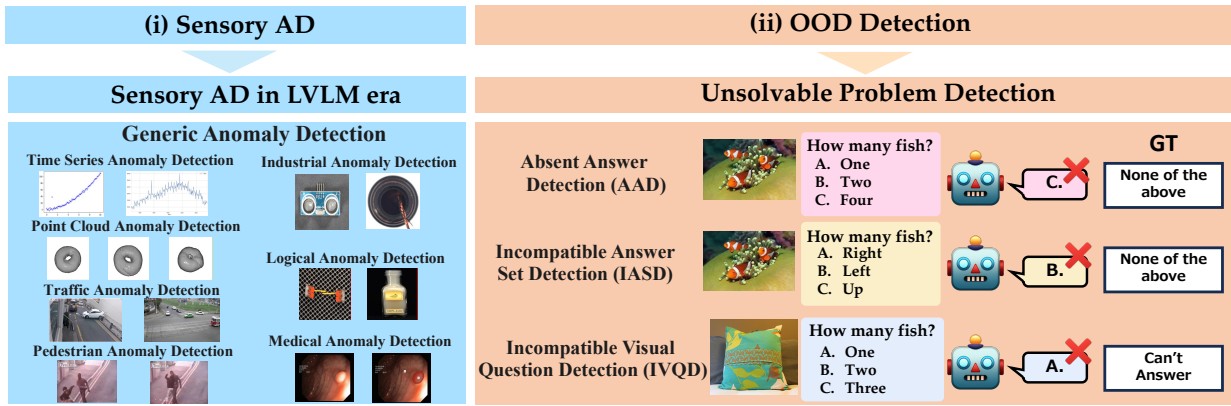

Figure 6: Overview of the evolution of each problem in the Large Vision Language Model (LVLM) era. (i) Sensory AD has consistently been an active research field in the LVLM era (Cao et al., 2023b). (ii) Notably, OOD detection is evolving into a new task called Unsolvable Problem Detection (Miyai et al., 2025a) in the LVLM era. Figure adapted partially from (Cao et al., 2023b) and from (Miyai et al., 2025a).

**Performance Rankings Vary Across Datasets**    From Table 3, we observe that the ranking of methods differs significantly depending on the dataset. For example, LoCoOp outperforms CoOp on the standard ImageNet far OOD benchmark, but underperforms on the near OOD benchmarks. This suggests that LoCoOp's OOD regularization strategy, which uses background OOD images during training, is especially effective for detecting semantically distant classes such as in far OOD settings, but not effective for near OOD settings. These results indicate the importance of evaluating OOD detection methods across multiple benchmarks, rather than relying on a single one.

**ImageNet-20 May Already Be Saturated**    On ImageNet-20, most methods achieve over 97% accuracy, making it difficult to distinguish performance differences among methods. Therefore, future research on OOD detection is likely to shift toward benchmarks with more diverse and larger numbers of classes.

**Room for Improvement Remains in Near and Hard-OOD Detection**    The results on ImageNet near OOD and ImageNet-X show that the methods with the highest score reach only around 80% accuracy. This indicates significant room for further improvement. We believe that future research in OOD detection will increasingly focus on these challenging benchmarks.

## 7    Evolution in LVLM Era

In this section, we introduce the early advances in OOD detection and AD in the Large Vision Language Models (LVLM) era. While previous sections focused on VLMs such as CLIP, this section shifts our focus to the more emerging topic of "Large" VLMs. Recent advancements in computer vision have led to the emergence of LVLMs such as GPT-4V (OpenAI et al., 2023) and LLaVA (Liu et al., 2024b). Although these fields are still in their early stages with limited papers, this survey provides a deep introduction to each problem in the hope that our detailed review can help foster further advancements in this area.

### 7.1    Change of Each Problem

**i. Sensory AD → Sensory AD**    Sensory AD has continued to develop in the LVLM era (Gu et al., 2024b; Cao et al., 2023b; Li et al., 2023b). The use of LVLMs has made AD applicable in many domains and modalities (Cao et al., 2023b).

**ii. OOD Detection → Unsolvable Problem Detection**    In the LVLM era, OOD detection has evolved into a new task termed Unsolvable Problem Detection (UPD) (Miyai et al., 2025a). UPD evaluates the LVLMs' ability to recognize and abstain from answering unexpected or unsolvable input questions, effectively expanding the scope of OOD detection into the context of Visual Question Answering (VQA) tasks. This

shift to the VQA task has significantly broadened the concept of OOD detection to a wider range of AI tasks involving LVLMs.

## 7.2 Unsolvable Problem Detection

### 7.2.1 Summary of Problem

**Background**   Following the recent revolutionary development of LLMs (Chiang et al., 2023; Touvron et al., 2023; Wei et al., 2023; Zhao et al., 2023), LVLMs (Awadalla et al., 2023; Dai et al., 2023; Liu et al., 2024b; Wang et al., 2023b; Ye et al., 2023; Li et al., 2023a; Lin et al., 2024) have demonstrated remarkable capabilities in diverse applications (Liu et al., 2024a;f; Yue et al., 2024). However, a significant concern has arisen regarding the reliability of these models, specifically their ability to generate accurate and trustworthy information. These models frequently produce incorrect or misleading information, a phenomenon referred to as "hallucination" (Bai et al., 2024b). Among the various hallucination issues (Bai et al., 2024b), the challenge of identifying out-of-place questions is crucial for deploying LVLMs in safety-critical applications. This challenge extends the concept of OOD detection to the VQA tasks for LVLMs and represents a specific aspect of LVLMs' trustworthiness.

**Definition**   Unsolvable Problem Detection (UPD) is a task to measure the trustworthiness of LVLMs, which is designed to evaluate models' capacity to withhold answers when faced with unsolvable problems. The UPD task can be categorized into three distinct problem types: Absent Answer Detection (AAD), Incompatible Answer Set Detection (IASD), and Incompatible Visual Question Detection (IVQD). Examples in each setting are shown in the right of Fig. 6. AAD evaluates the model's capacity to determine when the correct answer is absent from the provided options. IASD assesses the model's ability to discern answer choices that are completely irrelevant to the given question and image. IVQD evaluates the model's capacity to discern whether a question and image are unrelated or mismatched.

**Benchmark**   Miyai et al. (2025a) created MM-UPD Bench for the UPD challenge. MM-UPD encompasses MM-AAD, MM-IASD, and MM-IVQD benchmarks for each UPD problem. Each benchmarks are created on the top of MMBench (dev) (Liu et al., 2024f), which is a systematically designed objective benchmark for evaluating various abilities of LVLMs. Following the definition of each ability in MMBench (*e.g.*, "Coarse Perception: Image Scene" and "Logic Reasoning: Future Prediction"), MM-UPD evaluates the trustworthiness of LVLMs from various abilities.

Although MM-UPD is the main benchmark, the adaptation cost of creating UPD problems is not high, making it highly applicable to other benchmarks. For instance, the recently proposed MuirBench (Wang et al., 2025), a benchmark for multi-image understanding, has incorporated the concept of UPD by adding unsolvable problems.

**Application**   UPD has a wide range of applications, from everyday use of LVLMs to robot manipulation. Especially when incorporating LVLMs into safety-critical domains such as robot manipulation (Liu et al., 2024d) and autonomous driving (Li et al., 2024d), there is a risk of significant problems if the LVLM fails to identify erroneous user questions and makes incorrect predictions. UPD serves as a task to ensure safety in such safety-critical scenarios.

**Evaluation**   UPD employs three evaluation metrics: Standard Accuracy, which measures performance on solvable problems; UPD Accuracy, which assesses accuracy on unsolvable problems; and Dual Accuracy, which considers both aspects. Dual Accuracy considers a response correct only if the model successfully answers both the solvable and unsolvable problems within each paired set. The rationale is that ideal LVLMs should not only give correct answers for the solvable problems but need to withhold answering for unsolvable problems.

### 7.2.2 Findings

In the following, we briefly summarized the findings of the UPD challenge (Miyai et al., 2025a).

**1. Most LVLMs Hardly Hesitate to Answer.** Most LVLMs, especially open-source LVLMs, have significantly low UPD accuracies, which indicates the difficulty of the UPD challenge. For example, LLaVA-1.5 (Liu et al., 2024b) and CogVLM (Wang et al., 2023b), which are state-of-the-art LVLMs, completely fail to withhold answering. GPT-4V achieves higher performances than other LVLMs due to its safety training process (OpenAI, 2023). However, there is still a performance gap from the upper bound scores.

**2. Performance Tendency Differs a lot by Each Ability in the Benchmark.** The performance of LVLMs differs in each ability in the MM-UPD Bench. For instance, GPT-4V has its limitation in attribute comparison and LLaVA-NeXT-34B has its limitation in object localization.

**3. Effective Prompt Strategies Vary Across Different LVLMs** Effective prompt strategies vary across different LVLMs. In the original paper, they experimented with an option-based prompt approach that adds an option of "None of the above" and an instruction-based approach that adds an instruction "If all options are incorrect, answer None of the above". As a result, the effectiveness of each approach differs significantly depending on the type of LVLMs. This highlights the difficulty of finding an effective prompt strategy for all LVLMs.

### 7.3 Anomaly Detection in LVLM Era

#### 7.3.1 Summary of Problem

**Background** Anomaly detection is a crucial task in a variety of domains and data types. However, existing anomaly detection models are often designed for specific domains or modalities (Cao et al., 2023b). Also, current AD methods only provide an anomaly score for the test sample and require a manual threshold to distinguish between normal and anomalous instances for each sample (Gu et al., 2024b). To facilitate real-world applications, developing a system capable of expressing anomalies in natural language across various modalities and domains is crucial for ensuring accessibility to a wider range of users.

**Definition** The definition of AD remains consistent with conventional and VLM-based AD, aiming to identify samples that deviate from predefined normality. The key difference lies in the output format. Traditional methods generate an anomaly score as a numerical value, requiring a manual threshold to determine whether a sample is anomalous. In contrast, AD with LVLMs aims to recognize and describe anomalies using text, removing the need for manual thresholds and improving human interpretability.

**Benchmark** Since the field of AD with LVLMs is in its infant stage, there are still no unified benchmarks. AnomalyGPT (Gu et al., 2024b) focuses on industrial image anomaly detection/localization and uses the standard benchmarks MVTec-AD (Bergmann et al., 2019) and VisA (Zou et al., 2022). More recently, Cao et al. (2023b) extend the domain and modality and demonstrate the applications in industrial image anomaly detection/localization (*e.g.*, MVTec-AD (Bergmann et al., 2019)), point cloud anomaly detection (MVTec 3D (Bergmann et al., 2022b)), medical image anomaly detection/localization (*e.g.*, Chest X-ray (Kermany et al., 2018), Head CT (Felipe, 2018)), logical anomaly detection (*e.g.*, MVTec LOCO (Bergmann et al., 2022a)), pedestrian anomaly detection (*e.g.*, UCF-Crime Dataset (Sultani et al., 2018)), traffic anomaly detection (*e.g.*, Kaggle Accident Detection (Kay, 2018)), and time series anomaly detection (*e.g.*, Outlier Detection Dataset (User, 2021)).

**Evaluation** Evaluation in anomaly detection with LVLMs is an open challenge. AnomalyGPT (Gu et al., 2024b) asks LVLMs the question "Is there an anomaly in this image?" and determines anomaly or normal based on the simple rule-based approach of whether the response contains a "yes" or "no". However, this rule-based approach is not robust, as a response is considered correct even if the explanation following "yes" is completely incorrect. On the other hand, Cao et al. (2023b) conducted only qualitative evaluations and left quantitative evaluations as an open challenge. Therefore, the evaluation of anomaly detection by LVLM is a future challenge.

#### 7.3.2 Findings

Cao et al. (2023b) described the observations of GPT-4V in the paper, so we briefly summarized them here.

**1. GPT-4V Excels in Zero/One-shot Settings across Various Modalities and Fields.**    GPT-4V shows proficiency in identifying anomalies in multi-modality (*e.g.*, images, point clouds, X-rays) and multi-field (*e.g.*, industrial, medical, pedestrian, traffic, and time series anomaly detection). In addition, GPT-4V demonstrates strong performance in both zero-shot and one-shot settings.

**2. GPT-4V can Understand Both Global and Fine-grained Anomalies.**    GPT-4V can recognize both global and local abnormal patterns or behaviors, which indicates the ability to understand global and fine-grained semantics.

**3. GPT-4V can be Enhanced with Increasing Prompts.**    By giving more context and information, the model significantly improves its ability to detect anomalies accurately.

## 8   Future Directions

In this section, we discuss the future directions of OOD detection. We explore not only OOD detection for VLMs but also single-modal OOD detection, with a specific focus on emerging challenges as VLMs evolve. For a discussion of the long-standing challenges in OOD detection, we can refer the readers to the previous generalized OOD detection paper (Yang et al., 2024).

### 8.1   OOD Detection for Vision Language Models

**a. Hard OOD Detection**    Hard OOD detection will become increasingly important in the future due to its high practicality and the challenging nature of the problem. Hard OOD detection utilizes the OSR benchmark setup, where some classes within a single dataset are designated as ID and others as OOD. In this field, not only small datasets such as ImageNet-10 and ImageNet-20 (Ming et al., 2022a) but also datasets with a larger number of classes such as ImageNet-protocol (Palechor et al., 2023) and ImageNet-X (Noda et al., 2025) have been recently proposed. Many existing studies, such as LoCoOp (Miyai et al., 2023b) and LSN (Nie et al., 2023), primarily use the common ImageNet OOD benchmark, so hard OOD detection has not yet been well studied. This field will develop further in the future.

**b. Post-hoc Methods**    To propose post-hoc methods is important for the fundamental performance improvement of VLM-based OOD detection. The methods of directly employing an ID classifier such as MCM (Ming et al., 2022a) are called post-hoc methods. Prior to CLIP, various approaches were proposed (Hendrycks & Gimpel, 2017; Liang et al., 2018; Lee et al., 2018; Liu et al., 2020; Sastry & Oore, 2020; Sun & Li, 2022; Sun et al., 2022; Sastry & Oore, 2019). Nevertheless, VLM-based post-hoc methods often underperform methods with additional OOD prompts, so they are not extensively researched in zero-shot OOD detection. However, we should focus on the scalability of post-hoc methods. The post-hoc methods (Ming et al., 2022a; Miyai et al., 2025b) can be easily applied to many subsequent methods (Miyai et al., 2023b; Ming et al., 2022b; Nie et al., 2023; Li et al., 2024b), bringing fundamental performance improvements even for the few-shot setting. Furthermore, recently, post-hoc methods specifically tailored for prompt learning methods have also emerged (Jung et al., 2024). Therefore, proposing post-hoc methods and demonstrating the improvements not only in zero-shot but also in subsequent few-shot settings (Miyai et al., 2023b; Ming et al., 2022b) is crucial. This field should continue evolving, mirroring its growth before the advent of CLIP.

**c. Bridging the Gap with Closed-set Classifiers.**    OOD detection ensures the safety of ID classifiers, so it is crucial to bridge the gap between the advancements in existing closed-set classifiers and OOD detection. Currently, the representative method for few-shot OOD detection is LoCoOp (Miyai et al., 2023b), a text prompt learning method based on CoOp. However, in closed-set settings, VLM-based few-shot learning methods have been proposed other than CoOp (Chen et al., 2023a; Bulat & Tzimiropoulos, 2023; Lu et al., 2022). Therefore, adopting recent methods for OOD detection is essential for bridging the gap with closed-set ID classifiers.

**d. Training-free Few-shot OOD Detection**    The research direction of training-free few-shot OOD detection is still in its infant stage (Chen et al., 2024b; Zhang et al., 2024f). Considering the advancements of training-free methods in VLM-based AD, we anticipate a similar trajectory for VLM-based OOD detection.

Future directions include refining adapter-based methods or leveraging external knowledge such as retrieval augmentation (Udandarao et al., 2023; Ming & Li, 2024b). Addressing training-free few-shot OOD detection is a pivotal step towards realizing more computationally efficient OOD detection in the future.

**e. Full-spectrum OOD Detection**   VLM-based full-spectrum OOD (FS-OOD) detection is a promising research area (Lu et al., 2023; Yang et al., 2023). In practical applications, there is a strong motivation to create models that can not only detect semantically shifted OOD inputs but also generalize to covariate-shifted data (Yang et al., 2023; Bai et al., 2023). Within VLM-based methods, OOD detection and generalization are often discussed in separate contexts (Miyai et al., 2023b; Khattak et al., 2023), resulting in a trade-off between detection and generalization performance (Lafon et al., 2024). Therefore, we need to discuss both aspects together to mitigate the trade-off.

**f. Open-vocabulary OOD Detection**   Open-vocabulary OOD (OV-OOD) detection has a high practical potential, but it is still in its infant stages (Li et al., 2024b). The open-vocabulary setting has been actively explored in AD ( 5.1) and has the potential to become increasingly important in the field of OOD detection as well.

**g. Object-level OOD Detection**   Object-level OOD detection remains an unexplored area in VLM-based OOD detection. As discussed in Sec. 5.4, this is due to the vastness of the OOD space, which makes it difficult to identify OOD objects using texts. To pave the way for future advancements in object-level OOD detection/segmentation, foundation models for localization, such as SAM (Kirillov et al., 2023), offer a promising solution. By integrating these models with methods such as MCM (Ming et al., 2022a), we can potentially achieve object-level OOD detection and segmentation, opening up a new frontier in OOD detection research.

### 8.2   Single-modal OOD Detection

**a. Leveraging Large Pre-trained Models**   Leveraging large pre-trained models is important for single-modal OOD detection. Numerous methods for OOD detection conduct experiments using backbones trained from scratch and do not utilize pre-trained models (Yang et al., 2024; Zhang et al., 2024b; Hendrycks & Gimpel, 2017; Leys et al., 2018; Liu et al., 2020; Wang et al., 2022a; Sun & Li, 2022; Sun et al., 2022). In a recent study, Miyai et al. (2023a) systematically investigated the impact of pre-training on OOD detection from both the perspectives of the types of OOD data and pre-training algorithms (Chen et al., 2020; Caron et al., 2021). Dong et al. (2023) explored parameter-efficient learning for single-modal OOD detection and proposed DSGF, which leverages both fine-tuned features and original pre-trained features. While leveraging large pre-trained models (Dosovitskiy et al., 2021) with lightweight tuning is an active area of research in single-modal closed-set classification (Hu et al., 2022; Zhang et al., 2024e), there have been limited studies for single-modal OOD detection, which presents a promising avenue for future research.

**b. Real-world Benchmarks and Evaluations**   Considering the future development of VLM-based OOD detection, there should be increasing focus on expanding the scope of benchmarks to encompass real-world scenarios where CLIP is less applicable. For instance, recently, Baek et al. (2024) introduced ImageNet-ES, consisting of variations in environmental and camera sensor factors. Besides, utilizing datasets such as WILDS (Koh et al., 2021; Cultrera et al., 2023), which consider real-world data shifts, or datasets for medical OOD detection (Hong et al., 2024), can provide valuable insights, especially in safety-critical applications such as autonomous driving and medical image analysis.

## 9   Conclusion

In this survey, we comprehensively review the evolution of the five problems including AD, ND, OSR, OOD detection, and OD in the VLM era, and propose a framework of *generalized OOD detection v2*. Our framework identifies OOD detection and AD as the primary challenges in the VLM era. By articulating the shifts in the definitions, problem settings, benchmarks and methodologies, we encourage subsequent works to accurately understand their evolving target problems in the VLM era. By shedding light on recent studies in the LVLM era, we hope that researchers within each community can identify promising research directions in this

emerging era. By providing future directions, we hope that our survey will clarify the tasks to be tackled by future works in the VLM era, facilitating future advances in the right direction.

## Acknowledgment

This work was supported by JST BOOST, Japan Grant Number JPMJBS2418 and JST JPMJCR22U4. We thank Toyooka Mashiro (AYM Lab at UTokyo) for his valuable assistance in designing the figures, Kazuki Egashira, Yuki Imajuku, Takubon Son, and Zaiying Zhao (AYM Lab at UTokyo) for valuable feedback on the paper, and Shiho Noda for helping experiments.

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
