# OpenReview forum: "Generalized Out-of-Distribution Detection and Beyond in Vision Language Model Era: A Survey"
_TMLR — Accepted by TMLR_

### Review · Reviewer_LP7C · 2025-03-17

**Summary Of Contributions:**

This paper discusses how the field of anomaly/OOD detection tasks is shifting with the advent of VLMs (and touches on LVLMs). In particular, how different sub-problems have merged and other sub-problems become inactive in research. It also provides a comprehensive overview of research using VLMs for anomaly detection and OOD detection, followed by a brief discussion of the initial findings from work using LVLMs for the same problem.

**Audience:**

Yes

**Claims And Evidence:**

No

**Requested Changes:**

For acceptance, the authors should carefully consider what is the desired contribution of this work, and what are the key insights that are directly relevant to the introduction of VLMs in particular to this problem.

To strengthen the work, consider:

1. As the precise definitions for each of the five tasks are extremely important for the contributions of this work, it would have been better if the authors make the effort to carefully define each task here, rather than refer the reader to a previous work.

2. Although CLIP is certainly most common, I find the term "CLIP-based" to be inappropriate for a survey as the methods are not unique to CLIP, and other VLMs are applicable. "VLM-based" would be better.

3. There are grammatical errors throughout (e.g. OOD Detectin, TMRL2024, etc.)

**Strengths And Weaknesses:**

The question of how the emergence of VLMs impacts the field of AD/OOD detection is an important and timely one, and has not yet been directly targeted in this manner to the best of my knowledge. I appreciated the discussion about how the formulation of these problems have changed due to VLMs, however this is a relatively minor part of the paper. The literature survey of VLM-based AD/OOD detection is also solid. However, I have several issues with this paper.

Firstly, I do not agree with many of the findings and assertions. For example: it is highly questionable to call a sub-field "inactive" simply because it is not the direct focus of papers a from a handful of venues in the past 3 years. Also, the reasons given for this inactivity have nothing to do with the role of VLMs, rather for other phenomena such as "saturation in performance". The arguments for the merging of different problems is similarly lacking.

I also find the direction of the paper confusing at times. For example, in the VLM section, the focus is entirely on literature review. On the other hand, in the LVLM section, the authors choose to go beyond a literature review and give a detailed account of findings from a specific paper (Section 6.3.2), which is strange for a survey paper.

I also find the Future Directions section to be lacking in novel insights; most of the points are straight-forward summaries of the existing research directions.

---

> ### Author Response · Authors · 2025-04-23
> **Author Response to Reviewer LP7C**
>
> We sincerely thank you for your important feedback. We especially appreciate your recognition of the following strengths: "The question on the VLMs impacts is an important and timely one," "I appreciated  the discussion on the formulation of these problems in the VLMs era" and "The literature survey of VLM-based AD/OOD detection is also solid.”
>
> Below, we provide our responses to your concerns. The manuscript has also been updated accordingly, with the changes marked in red.
>
> *※ Unless otherwise noted, references to section or figure numbers refer to those in the original version of the manuscript.*
>
> > ### **W1: Justification of the inactivity**
>
> Thank you very much for your thoughtful feedback. We would like to clarify that our use of the term “inactive” was solely intended to refer to the noticeable decrease in the number of publications in major venues following the emergence of CLIP. It was not our intention to suggest that the field has come to an end as we noted in Section 2.4. If the term “inactive” conveyed an unintended negative impression, we sincerely apologize and are fully open to revising the wording to a more appropriate and respectful alternative.
>
> Regarding the performance saturation, we believe it is related to VLMs. The adoption of powerful VLMs has often led to significantly improved performance compared to earlier methods without VLMs, which we regard as part of a paradigm shift driven by VLMs.  As for the merging of OSR and OOD detection, as discussed in Section 2.3, we consider the benchmark setup that originally defined OSR to have been effectively incorporated into the hard OOD detection. Before the VLM era, the subtle differences between the two fields often caused confusion, as similar methods were proposed for each setting. Given the recent decline in papers on OSR and their identical definitions, we believe it is more reasonable to treat OSR as a part of OOD detection, in order to help the community avoid any confusion.
>
> > ### **W2: The section on LVLMs mainly summarizes findings from individual papers.**
>
> Thank you for your comment. As the research on LVLMs in this field is still in the early stages of development and remains less mature than that of VLMs, we did not attempt to provide a comprehensive summary in this section. Instead, we briefly introduced individual papers to give readers an overview of the current landscape. We included this section to encourage further research and progress in this emerging area. We believe that a more in-depth analysis is better suited for future work.
>
> >  ### **W3: Future directions lacks novelty.**
>
> Thank you for your comment. We would like to emphasize that the proposed future directions are not merely a summary of existing studies, but rather reflect novel insights derived from our comprehensive survey. For example, our discussions on training-free few-shot OOD detection, open-vocabulary OOD detection, and object-level OOD detection are motivated by the observation that these areas are actively explored in anomaly detection but remain underdeveloped in the context of OOD detection. These directions emerge from the comparative analysis presented in this survey and would not have been identified by simply summarizing prior work.
>
> > ### **RC1:  Include the definition of each task in this paper**
>
> We appreciate your comment. In Section 2.1, we have explicitly provided definitions for the five tasks in the updated version. These definitions are based on the framework of Generalized OOD Detection v1, as they are currently the most widely accepted definitions within the research community. By clearly explaining these definitions, we aim to help readers understand and ensure conceps of these tasks within this paper.
>
> > ### **RC2: "VLM-based" would be better.**
>
>  Thank you for your insightful comment. We agree that the scope should not be limited to CLIP. Therefore we have updated the term to "VLM-based methods". However, since not all VLMs use softmax scoring (e.g., SigLIP [1] uses sigmoid-based scores for the classification), methods such as MCM or GL-MCM cannot be directly applied to such VLMs. We have therefore added a clarification noting that while we adopt the broader term "VLM-based methods," CLIP remains dominant and not all VLMs are universally applicable.
>
> Reference
>
> [1] Zhai+, Sigmoid Loss for Language Image Pre-Training, ICCV2023
>
> >  ### **RC3:  Fix grammatical errors**
>
>  Thank you for pointing that out. We have corrected the grammatical errors in the points you pointed out and other points we found.

---

> > ### Comment · Reviewer_LP7C · 2025-05-22
> >
> > Thank you for your response.
> >
> > I still have some concerns about W1. I find this section a little troubling because it makes sweeping statements about entire sub-field based on work from a handful of conferences over a couple of years. This criteria for inactivity has obvious flaws, and even this criteria is not rigorously maintained; for example: OD -> Inactive, yet ICLR2024 in this area and "it may be active again in the future". To me this is not a very meaningful or convincing statement. Furthermore, I find some claims to be questionable. For example "Sensory AD -> VLM-based AD", and "... sensory AD has become a highly active and noteworthy field in the VLM era". Sensory AD has always been a very active field even before VLMs, and there remains substantial research in this area that does not use VLMs. Many of these methods achieve arguably saturated performance, e.g. [1], so I disagree with the argument that VLMs caused this.
> >
> > [1] Roth, Karsten, et al. "Towards total recall in industrial anomaly detection." Proceedings of the IEEE/CVF conference on computer vision and pattern recognition. 2022.

---

### Review · Reviewer_sHxj · 2025-04-04

**Summary Of Contributions:**

This paper provides a survey on recent contributions on out-of-distribution detection (OOd), especially concerning vision-language models (VLMs). The authors emphasize how the out-of-distribution litterature has evolved in the _era_ of VLMs, where some fields have evolved into new problems, and some went inactive. The authors then summarize OOD problems in the VLM era, being divided mainly into CLIP-based OOD and AD.

**Audience:**

Yes

**Claims And Evidence:**

Yes

**Requested Changes:**

Here is a list of requested changes before I can recommend acceptance,

- The authors should make the effort of making pedagogical illustrations for the different methodologies they approach. For instance, the authors could include an illustration for each case in Table 2, or an overall illustration highlighting the differences of each setting. The authors can take for example Fig. 4 which I think is very direct and clear in showing the difference between OOD with/without prompting

- Include benchmarking/evaluation with the most representative methodologies on widely used benchmarks

- For enhancing the originality of their work, the authors should: i) rewrite the summary of their work, and ii) re-design figures 1 and 5.

**Strengths And Weaknesses:**

__Strengths__

- The paper highlights different trends and the overall evolution of OOD with VLMs. In my view, the trend that mostly comes out of the paper is that CLIP is the _de facto_ representation learning backbone for OOD with VLMs.

- The authors do a good job summarizing emerging problems in the area.

__Weaknesses__

- The authors do not provide any insights on empirical aspects of the area at hand. For instance, a comparison between at least a subset of the discussed methods (e.g., the most relevant) would be helpful to other researchers.

- While the paper is comprehensive, the paper is not very pedagogical in its nature. For instance, in all the paper, there is only a single illustration involving the different methodologies discussed (i.e., Fig 4).

- The authors re-use contents of other published papers. While the authors do cite the sources correctly, I think the authors could use the chance of explaining the subject in their own words and style. For instance, figures 1 and 5 lack originality and are way too similar to previous works. The same can be said about the introduction in comparison with (Yang et al., 2024).

---

> ### Author Response · Authors · 2025-04-23
> **Author Response to Reviewer sHxj**
>
> We sincerely thank you for your constructive feedback.  In particular, we are deeply grateful for your positive comment: "The authors do a good job summarizing emerging problems in the area.”
>
> Below, we provide our responses to your concerns. The manuscript has also been updated accordingly, with the changes marked in green.
>
> *※ Unless otherwise noted, references to section or figure numbers refer to those in the original version of the manuscript.*
>
>
> > ### **RC1, W2: Add illustrations of the methodologies**
>
>  We appreciate your thoughtful suggestion. We agree that adding a methodological figure of the approaches can significantly enhance the reader's understanding. Following your advice, we have updated Figure 5 in the revised version and include illustrations about each training type in both zero-shot and few-shot settings.
>
> > ### **RC2, W1: Include benchmarking/evaluation results**
>
>   We sincerely appreciate your valuable feedback. In the updated version, we have incorporated an experimental section (Section 6) to complement the main discussion. Given that this work is intended to provide a deeper understanding of OOD detection, the experiments are deliberately scoped within this context.
>
> The experiments yield three key insights: (1) **Performance rankings vary across datasets**, (2) **ImageNet-20 may already be saturated**, and (3) **There remains significant room for improvement in large-scale near and hard-OOD detection**.
>
> These findings highlight the importance of developing and evaluating future methods on more challenging and diverse benchmarks, especially those that involve hard OOD detection in large-scale classification settings.
>
> > ### **RC3, W3: Improve the originality of this work**
>
>  Thank you for your valuable feedback. Following your advice, (i) we revised the summary in the Introduction, and (ii) redesigned Fig. 1 and Fig. 5 (Fig. 6 in the updated version). For the framework figure of Generalized OOD Detection in Fig. 1, we consider that the structure of the original paper’s figure provided the clearest explanation. While we referred to its structure, we introduced originality by modifying the color scheme, fonts, and overall design to maintain clarity while enhancing readability.

---

### Review · Reviewer_KpyY · 2025-04-16

**Summary Of Contributions:**

They introduce an updated unified framework termed generalized OOD detection v2, extending previous framework with inclusion of some works in the era of VLMs and LVLMs. They also identify some interesting future directions.

**Audience:**

Yes

**Broader Impact Concerns:**

No particular concern.

**Claims And Evidence:**

No

**Requested Changes:**

1. When introducing similar tasks in the second paragraph of Intro, the task definition should be provided to help readers understand how they are different from each other. These are important concepts to make clear since the start of the paper. I’d suggest shift some part from Sec 2.1 upfront to make the reading smoother.
2. Fig 1 caption should be enriched to be self-contained, with all concepts clearly explained.
3. The OOD discussed in this paper only covers papers working on image data. OOD on videos is another major line of works, for instance, on video anomaly detection (VAD), but is completely missed in this survey.  Leveraging VLMs or LVLMs in VAD is an active research field. I’d suggest to include this line of research to make the survey comprehensive. Some references are listed below but not exhaustive [1,2,3] just to provide a pointer. The inclusion might also impact the conclusion of the paper. For instance, VAD is likely categorized under semantic AD, which is quite active.
4. In general, update related work discussion to include more recent works appeared in 2025. For instance, some accepted CVPR 2025 papers are available publicly? Also, update Table 1 and add references to each paper.
5. Sec 6 is the most interesting part to me, as it covers the recent advances and emerging trends. Unfortunately I found this section too slim: 1) when drawing conclusions/findings, there are no explicit experimental proof to validate the findings. 2) The findings seem all from prior works, what are the novel findings?

=====
[1] Zanella, Luca, et al. "Harnessing large language models for training-free video anomaly detection." CVPR, 2024.
[2] Zanella, Luca, et al. "Delving into clip latent space for video anomaly recognition." Computer Vision and Image Understanding 249 (2024): 104163.
[3] Zhang, Huaxin, et al. "Holmes-vau: Towards long-term video anomaly understanding at any granularity." CVPR, 2025

**Strengths And Weaknesses:**

The taxonomy organization is nicely structured with clear definition.

Some major weakness are listed below:

1. Some important concepts about relevant OOD tasks should be put clear earlier in the survey, to make the flow clear.

2. Papers covering video-based OOD task, like video anomaly detection, a quite active line of works, are missing in the survey. By including them will enrich the survey and might slightly change the conclusions.

3. The content in LVLM part is a bit slim, lacking novel findings and experimental proof.

Detailed requested changes are listed below in detail.

---

> ### Author Response · Authors · 2025-04-23
> **Author Response to Reviewer KpyY**
>
> We sincerely appreciate your thoughtful review, which was very helpful in improving the readability of our survey.  In particular, we are deeply grateful for your positive comment: “The taxonomy organization is nicely structured with clear definition.”
>
> Below, we provide our responses to your concerns. The manuscript has also been updated accordingly, with the changes marked in magenta.
>
> ※ *Unless otherwise noted, references to section or figure numbers refer to those in the original version of the manuscript.*
>
> > ### **RC1, W1: Clarify task definitions earlier in the Introduction.**
>
> Thank you for your valuable suggestion. We agree that introducing the task definitions earlier in the introduction is important for enhancing the reader’s understanding. In the updated version, we have followed your advice and moved the task definitions in the introduction to improve the overall readability.
>
>
> > ### **RC2: Enrich the captioning in Fig. 1**
>
> Thank you for your feedback. We have revised the caption of Fig. 1 to make it more informative and self-contained.
>
> > ### **RC3, W2: Discussion on video anomaly detection**
>
> Thank you for your valuable comment. We acknowledge that video anomaly detection (VAD)  is an actively studied and important research area. However, as of now, there is limited work specifically addressing OOD detection in video settings using VLMs.  As stated in Section 2.2, the goal of this survey is to deepen the understanding of OOD detection while referring to related fields such as AD as supplementary context. Therefore, we did not conduct a comprehensive review of VAD in this work. However, based on the reviewer’s helpful feedback, we agree that including even a brief section on VAD rather than omitting it entirely can better clarify the distinction between OOD detection and AD. Therefore, we have added this additional discussion in the updated Section 5.3.3.
>
>
> > ### **RC4: Include more recent papers such as CVPR2025**
>
> Thank you for your suggestion. We have added more recent papers from ICLR 2025 and CVPR 2025 to Tables 1 and 2, and have added the references in Table 1. While several papers on OOD detection have been accepted to CVPR 2025, they are not yet available on the official site and arXiv. We plan to include them in the final version of the paper if they are made publicly available by the time of final version submission.
>
> > ### **RC5, W3: The content in the section on LVLMs is short.**
>
> Thank you for your comment. As the research on LVLMs in this field is still in its early stages and remains less mature than that of VLMs, we chose to refer to the insights of individual papers in that section rather than present new experimental findings. We believe that a more comprehensive exploration of this area is better suited for future work.

---

### Decision · Action_Editor_d1yG · 2025-06-04

**Recommendation:** Accept with minor revision

**Comment:**

This paper presents a comprehensive survey of image-based out-of-distribution detection (and related fields) in the context of the development of vision language models. It is timely and provides guidance for promising research directions in the area. Overall it is a good contribution to the literature and I recommend acceptance. However, claims and methodology used in sections on the evolution of the field (2.2, 2.3) should be revised as below in the final version of the paper.

Minor changes to be made are listed below:
- The definition of a 'top venue' used and if possible, how the search in Section 2.2 was done would provide better clarity on the methodology of how research activity was defined. Also clarify the time period being considered in the main text.
- Adjust statements to address [reviewer LP7C's comments](https://openreview.net/forum?id=FO3IA4lUEY&noteId=PqU3tyDZ00)

**Audience:**

Out-of-distribution detection is a core research topic in AI and this survey in the context of the current vision-language model era will certainly be of interest to the TMLR audience.

**Claims And Evidence:**

The paper is primarily a survey of out-of-distribution detection and related fields given the recent developments in vision language models. Reviewers agree it comprehensively covers the literature in the image domain and provides interesting future directions. Some claims about the evolution of the field are less well supported and these should be adjusted as detailed in the comments; these however do not affect the central contribution of the work.